# From Histopathology Images to Cell Clouds: Learning Slide Representations with Hierarchical Cell Transformer

## Abstract

It is clinically crucial and potentially beneficial to analyze and directly model the spatial distributions of cells in histopathology whole slide images (WSI). However, existing methods typically analyze WSIs via image representation learning and ignore the importance of cell distributions. Thus, it remains an open question whether deep learning models can directly and effectively analyze WSIs from the semantic aspect of cell distributions. In this work, we argue that each WSI can be regarded as a collection of cells and propose a new scheme consisting of cell detection and cell cloud modeling to tackle these challenges. Firstly, we propose a novel human-in-the-loop label refinement method to finetune the pretrained cell detection and classification model. Then, a novel hierarchical **Cell Cloud Transformer (CCFormer)** is proposed to model the cell spatial distribution. Specifically, a Neighboring Information Embedding module is proposed to characterize the distribution of cells within the cell neighborhood, and a Hierarchical Spatial Perception module is proposed to learn the spatial relationship among cells in a bottom-up manner. Clinical analysis indicates that clinical evaluation metrics directly based on counting cells can effectively assess patients' survival risk, offering significant potential for analyzing and modeling cell distribution in WSIs. Besides, extensive experiments on survival prediction and cancer staging show that CCFormer achieves state-of-the-art performances and evidently outperforms other competing methods by learning from cell spatial distribution alone. Our project is available at anonymous link.

## 1 Introduction

Analyzing histopathology whole slide images (WSIs) presents a significant challenge in computational pathology. It requires managing gigapixel images while capturing the features and distributions of tissues and cells. Significant progress in WSI analysis and related downstream tasks has been achieved by training models on high-quality WSI datasets, such as those from the Cancer Genome Atlas Program (TCGA) (Liu et al., 2018). These advancements include tasks like survival prediction (Chen et al., 2021; Shao et al., 2024), cancer staging (Li et al., 2024; Qiu et al., 2025), cancer sub-typing (Song et al., 2024), and gene mutation prediction (Xu et al., 2024).

Existing methods (Ilse et al., 2018; Chen et al., 2021; 2022) typically analyze WSIs via conventional image perception frameworks, where the image representation is the cornerstone of downstream tasks. Thus, numerous histological foundation models (Xu et al., 2024; Wang et al., 2024; Chen et al., 2024; Lu et al., 2024a; 2023) have been proposed, pre-trained on large-scale datasets for general-purpose representations. Heavy reliance on the foundation models results in high computational costs. Unlike natural images, the analysis of *cell spatial distribution* within WSIs has been verified as clinically important, associated with the molecular profile (Saltz et al., 2018), tumor progression (Corredor et al., 2019), prognostic biomarkers (Page et al., 2023), *etc.* However, analyzing WSIs by modeling the cell spatial distribution has been overlooked and remains an open problem in the deep learning community.

We argue that a WSI can be regarded as a cell cloud, and explicitly modeling cell distribution from the cell cloud can provide better slide representation from the WSI. As shown in Figure 1, compared

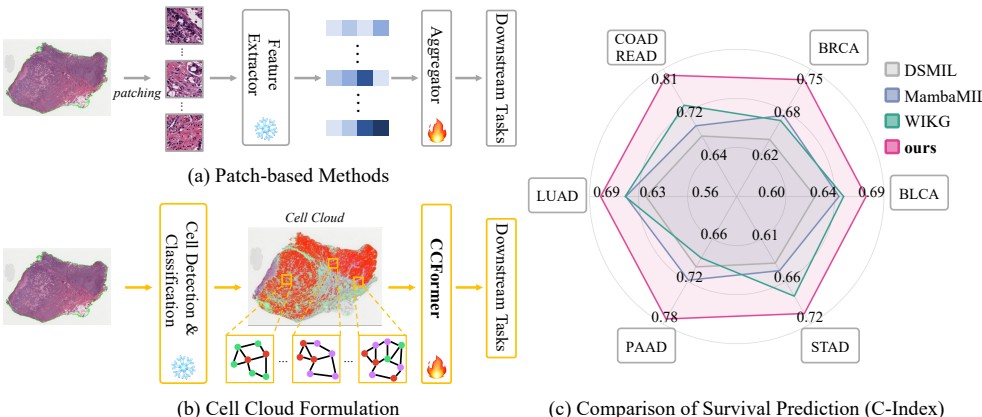

Figure 1: Patch-based methods extract patch features using pre-trained feature extractors and obtain slide-level representations by trainable aggregators, while Cell Cloud Transformer (CCFormer) directly learns the cell spatial distribution. CCFormer outperforms SOTA patch-based methods across multiple cancers.

with patch-based methods (Ilse et al., 2018; Lu et al., 2021; Li et al., 2021; Shao et al., 2021; Chen et al., 2021; Li et al., 2024), which aggregate multiple patch features to form a slide representation, our cell-cloud approach can model the microscopic topological relationships of cells. This cell cloud formulation can provide better microenvironment modeling and interpretability for clinical analysis. For example, cell differentiation and aggregation status affect clinical prognosis (Saltz et al., 2018; Corredor et al., 2019; Page et al., 2023; Diao et al., 2021).

To achieve the above goals, we propose a new scheme consisting of cell detection and cell cloud modeling. First, we pretrain a network to detect and classify cells within WSIs and propose a novel Human-in-the-Loop Label Refinement (HLLR) to reduce the costs of high-quality cell labeling within WSIs in domain-adaptation finetuning. Second, we propose a novel hierarchical cell cloud Transformer, termed CCFormer, to model cell clouds.

The cell cloud exhibits a significant hierarchical structure, which can be mapped to various kinds of clinical concepts (Diao et al., 2021): local cell clusters (.$e.g$, tumor cellularity), larger cell spatial distribution structures ($e.g.$, cancer-associated stroma), and the tissue microenvironment at the WSI level that can reflect clinical indicators such as cancer stage and patient survival risk. This motivates us to design the CCFormer, which consists of two key modules: **Neighboring Information Embedding (NIE)** and **Hierarchical Spatial Perception (HSP)**. NIE describes the neighborhood cell distribution pattern of cells at the cell level by evaluating the statistical characteristics of each type of cell within the cell neighborhood. HSP further progressively perceives and aggregates cell spatial distribution information hierarchically. The clinical analysis based on cell clouds indicates that the survival risk of patients can be effectively decided by evaluating the proportions of various cell types, which is difficult to obtain based solely on WSI. Extensive experiments on survival prediction and cancer staging show that analyzing WSIs via cell clouds is a highly competitive framework. Our contributions can be summarized as follows:

- **Efficient WSI Analysis Based on Cell Clouds**: We introduce an innovative approach for analyzing WSIs that directly and effectively models the pathological microenvironment through the semantic lens of cell clouds and propose a novel CCFormer to explicitly capture and learn the intricate microscopic topological relationships among cells.

- **Low-Cost and High-Quality Domain-Adaptation Finetuning**: We propose a novel Human-in-the-Loop Label Refinement that substantially reduces the cost of manual cell-level annotation during domain-adaptation finetuning, further yielding better cell detection and classification results for downstream clinical endpoints.

- **Hierarchical Cell Spatial Distribution Representation**: We introduce a novel Neighboring Information Embedding (NIE) technique to capture neighborhood cell distribution at

the cell level, along with a Hierarchical Spatial Perception (HSP) method to model cell spatial distribution information in a bottom-up manner.

- **Extensive Experimental Validation**: Clinical analysis confirms that cell clouds can be directly utilized to develop effective clinical indicators. Furthermore, extensive experiments demonstrate the efficacy of the cell cloud framework and the CCFormer model.

## 2    RELATED WORK

**Patch-Level Methods in Histopathology.** Patch-level methods (Ilse et al., 2018; Shao et al., 2021; Chen et al., 2021; Li et al., 2024; Shao et al., 2024; Chan et al., 2023; Lin et al., 2023; Wu et al., 2025; Zhang et al., 2025; Dong et al., 2025; Tang et al., 2024) divide WSIs into patches and employs pre-trained models (He et al., 2016; Lu et al., 2024b; Chen et al., 2024; Lin et al., 2023) to extract patch features for downstream tasks. Since WSIs are typically giga-pixel images, most existing methods (Ilse et al., 2018; Shao et al., 2021) are designed with Multi-Instance Learning (MIL), where WSIs are formulated as a bag of sampled patch features. Although MIL-based methods can effectively analyze WSIs, these methods only focus on sampled regions of interest, limiting to learning the spatial and semantic relationship of patches across the WSI. To track this issue, graph of patches has been introduced into WSI analysis (Chen et al., 2021; Li et al., 2024; Shao et al., 2024; Chan et al., 2023; Shi et al., 2024). Patch-GCN (Chen et al., 2021) introduces patch-based graph convolutional networks to model the relationship among patches. Although graph-based methods can describe the relationships among patches, they are limited at the patch-level and unable to model the cell spatial distribution. In addition, Ceograph (Wang et al., 2023) propose to analyze cell spatial organization with graphs. However, Ceograph focuses on learning cell relationships within each patch and can not percept the cell distribution across the WSI. In this paper, we formulate WSIs as cell clouds and propose to model cell spatial distribution across the entire WSI.

**Point Set Learning.** Point set learning aims to understand the spatial relationships between points in point sets, also known as point clouds. Recently, deep learning approaches for learning point clouds have been rapidly developed and can be categorized into projection-based (Su et al., 2015; Lang et al., 2019), voxel-based (Maturana & Scherer, 2015; Choy et al., 2019; Graham et al., 2018; Chen et al., 2023), point-based (Qi et al., 2017a;b; Ma et al., 2022; Zhao et al., 2019; Wu et al., 2022), and serialized methods (Wu et al., 2024; Wang, 2023; Liang et al., 2024). Since projection-based and voxel-based methods are typically designed for 3D point clouds, these methods are difficult to apply to 2D cell clouds. While point-based methods can be easily extended to 2D cell clouds, existing methods are not suitable for describing the unique hierarchical spatial relationships among cells. In addition, serialized methods organize points into sequences based on predefined patterns and lack flexibility in handling the varied hierarchical structures of cell clouds. In this paper, we propose CCFormer, which progressively learns the relationships among cells hierarchically.

## 3    PILOT STUDY - CLINICAL ANALYSIS

Cell-level annotations and statistics hold significant clinical importance (Corredor et al., 2019; Diao et al., 2021; Saltz et al., 2018). As a pilot study, we construct survival risk evaluation metrics based on cell clouds and conduct Kaplan-Meier analyses. Based on clinical experience, the survival risk is highly influenced by the proportion and distribution of neoplastic and inflammatory cells (Page et al., 2023). Therefore, we construct the Cell Proportion Score (CPS) , which considers the proportions of various cell types within the WSI:

$$S_{CPS} = [\frac{N_{neo}}{N_{total}}, 1 - \frac{N_{inf}}{N_{total}}, \frac{N_{neo}}{N_{inf}}]\boldsymbol{\alpha}, \tag{1}$$

where $S_{CPS}$ denotes Cell Proportion Score, $\boldsymbol{\alpha}$ is the weight vector, and $N_{total}$, $N_{neo}$, $N_{inf}$ are the number of cells, neoplastic cells, and inflammatory cells within a WSI, respectively. For different types of cancer, $\boldsymbol{\alpha}$ can be set based on the type of cancer to focus on different components. Pilot studies on TCGA-COADREAD and TCGA-PAAD show that CPS correctly distinguished between high-risk and low-risk patients with log-rank p-values of 1.54e-03 and 1.43e-03, respectively.

**Conclusion.** The pilot study shows that simple cell-level statistics can be utilized to develop effective clinical indicators, providing strong support for cell cloud–based WSI representations. Building

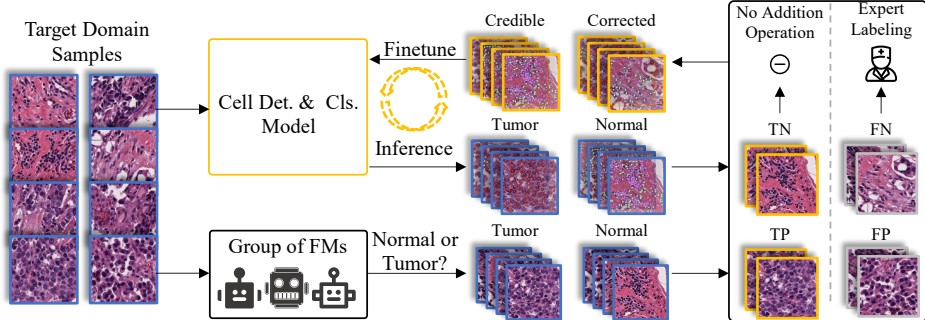

Figure 2: **Human-in-the-Loop Label Refinement (HLLR).** HLLR reduces the annotation cost for domain-adaptation finetuning by integrating foundation model-based sample selection and expert labeling of hard samples.

on this, we propose CCFormer to extend cell count–based statistics into hierarchical modeling of cell spatial distributions, enabling finer-grained learning of topological relationships among cells.

## 4 CCFORMER

The collection of cells within a WSI is detected and classified via a pretrained model. To generate better cell clouds for analyzing WSIs, we propose the Human-in-the-Loop Label Refinement (Figure 2) to generate samples for domain-adaptation finetuning of the pretrained cell detection and classification model. Then, we propose a hierarchical Cell Cloud Transformer (Figure 4) to model the cell spatial distribution and further apply it to clinical endpoints.

### 4.1 HUMAN-IN-THE-LOOP LABEL REFINEMENT

We perform preliminary annotations on WSIs with a model pretrained on PanNuke (Gamper et al., 2020). Due to the differences in data distribution between PanNuke and the target dataset, such as TCGA, domain-adaptation finetuning on the target dataset is helpful for better cell classification. Finetuning on a large-scale human-annotated dataset incurs substantial annotation cost (Hörst et al., 2025). Therefore, as shown in Figure 2, we propose a Human-in-the-Loop Label Refinement (HLLR) that combines foundation-model-based high-confidence screening with expert correction of hard samples. First, patches whose foundation-model judgments agree with cell-based assessments are treated as credible and are directly used to finetune the cell detection and classification model. Specifically, foundation-model decisions are obtained via voting across multiple models (Lu et al., 2024a; Ikezogwo et al., 2024; Sun et al., 2023), whereas the cell-based assessment is computed from the predicted proportion of cancer cells within the patch. Second, for the remaining patches, we sample a subset for manual correction and include the corrected patches in the refinement set used for finetuning. HLLR markedly reduces manual effort while maintaining effective adaptation to the target WSIs. By finetuning on the dataset of credible patches and expert-corrected difficult patches, the pretrained model captures cell morphology in the target domain, yielding better cell detection and classification results for downstream high-level pathology tasks.

### 4.2 HIERARCHICAL CELL CLOUD TRANSFORMER

**Neighboring Information Embedding.** The neighborhood cell distribution patterns at the cell level are critical characteristics in distinguishing the cells within the same category. For cells similarly labeled as cancer, whether they are surrounded by a large number of cancer cells or immune cells have completely different clinical significance (Wang et al., 2023). Thus, we propose NIE to embed the spatial distribution information of neighboring cells. Specifically, we propose local and global density features to embed statistical information of neighboring cells.

For each WSI, the mean shortest distance among cells $d_{mean}$ across the dataset is used to adaptively set the largest local neighborhood radius $r_{max} = \lambda_r d_{mean}$, where $\lambda_r$ is the scale factor. To ob-

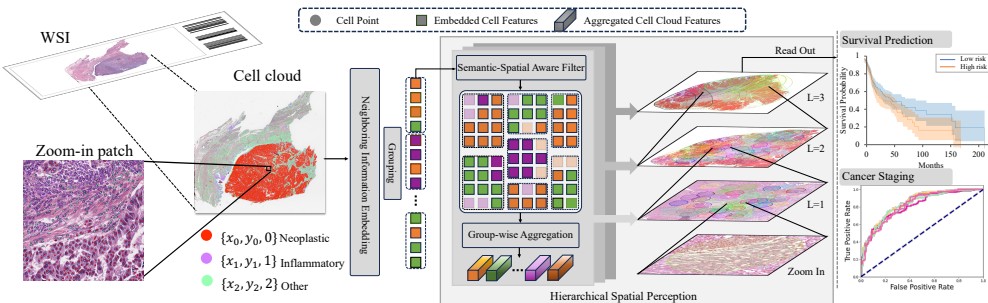

Figure 4: **The pipeline and illustration of CCFormer.** Given the cell point (coordinate and type) within the cell cloud, Neighboring Information Embedding supplements the statistical characteristics of its neighboring cells. Hierarchical Spatial Perception further progressively perceives and aggregates cell spatial distribution information hierarchically. Finally, the feature of cell spatial distributions across the entire WSI is applied to clinical endpoints.

tain more precise local spatial information, we introduce a discrete number $N_d$ to uniformly divide $r_{max}$ into multiple segments, thereby obtaining a series of radius $\mathbf{r} = [r^{(1)}, r^{(2)}, \cdots r^{(N_d)}]^T$. We denote the number of the $t$-th type of cell within the $j$-th radius of the $i$-th cell as $N^{(i,r^{(j)},t)}, i \in \{1, 2, \cdots, C\}, j \in \{0, 1, 2, \cdots, N_d\}, t \in \{1, 2, \cdots, T\}$, where $C$ is the number of cells and $T$ is the number of cell types. Specifically, $N^{(i,r^{(0)},t)} = 0$. Thus, the local relative density feature is computed as follows:

$$f_{ld}^{(i,r^{(j)},t)} = \frac{N^{(i,r^{(j)},t)} - N^{(i,r^{(j-1)},t)}}{N^{(i,r^{(N_d)},t)}}, \tag{2}$$

where $f_{ld}^{(*)}$ denotes the local density feature of the $t$-th type of cell within the $j$-th radius of the $i$-th cell. The local density feature vector of $i$-th cell $F_{ld}^{(i)} = [f_{ld}^{(i,r^{(1)},1)}, \cdots f_{ld}^{(i,r^{(N_d)},T)}]^T$. Equation 2 measures the relative density of cells within multiple neighborhood radii, thereby quantifying the proximity between cells and their neighboring cells.

We further introduce the global density feature to quantify the statistical distribution of cells across the entire cell cloud:

$$f_{gd}^{(i,r^{(j)},t)} = \frac{N^{(i,r^{(j)},t)} - N^{(i,r^{(j-1)},t)}}{\max_{i \in \{1,2,\cdots,C\}}(N^{(i,r^{(N_d)},t)})}, \tag{3}$$

where $f_{gd}^{(*)}$ denotes the global density feature of the $t$-th type of cell within the $j$-th radius of the $i$-th cell. The global density feature vector of $i$-th cell $F_{gd}^{(i)} = [f_{gd}^{(i,r^{(1)},1)}, \cdots f_{gd}^{(i,r^{(N_d)},T)}]^T$. In CCFormer, the embedding feature of each cell $F_{cell} \in \mathbb{R}^{T+2N_dT}$ is the concatenation of $F_{ld} \in \mathbb{R}^{N_dT}$, $F_{gd} \in \mathbb{R}^{N_dT}$, and the one-hot encoding of cell type.

We generate toy point sets with the Gaussian distribution to illustrate NIE. As shown in Figure 3, NIE not only correctly distinguishes points of different categories but also further differentiates points of the same type located in different neighborhood patterns. In Section 5,

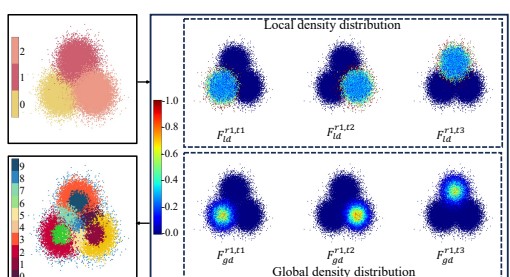

Figure 3: **Toy example of NIE.** A toy point set containing three categories is generated. After extracting features via NIE, we performed K-Means clustering. The results indicate that features derived from NIE can effectively differentiate points at different locations (boundaries, core regions, and outliers).

we further validate that the NIE can also effectively describe the neighborhood cell distribution on real cell clouds.

**Hierarchical Spatial Perception.** In WSI, the spatial distribution of cells is hierarchical. The entire WSI is composed of multiple important regions, each of which consists of smaller clusters of

---

**Algorithm 1** Hierarchical Spatial Perception (HSP)

---

IUPUT: Coordinates of cells $C = \{c^{(i)}\}_{i=1}^{N_{total}}$, features of cells $F_{cell} = \{f_{cell}^{(i)}\}_{i=1}^{N_{total}}$.
PARAMETER: The number of group anchors $N_k$, the filter threshold $\lambda_{sim}$, the basic number of points within each group $N_{basic}$, the number of perception levels $L$.

1: Encode features $F = \mathcal{F}_{fc}(F_{cell})$, where $\mathcal{F}_{fc}$ is a fully connected layer.
2: **for** level $l = 1, \cdots, L$ **do**
3:     Generate coordinate of group anchors $N_k$ by FPS.
4:     Group points according to $C$ and $N_k$.
5:     **for** group $k = 1, \cdots, N_k$ **do**
6:         Compute $S_{sim}$ according to Equation 4.
7:         Create $M = bool(S_{sim} > \lambda_{sim})$.
8:         Update $F$ according to Equation 5.
9:         Compute spatial distribution of $k$-th group $f_{group}^{(k)}$ with mean aggregation.
10:     **end for**
11:     $C \leftarrow C_k$.
12:     $F \leftarrow \{f_{group}^{(k)}\}_{k=1}^{N_k}$
13:     $N_k \leftarrow N_k/N_{basic}$
14: **end for**
15: **return** Slide representation $f_{cell}^{(WSI)} = \text{MaxAgg}(F)$.

---

cells. We propose HSP to learn this hierarchical structure of cell clouds. Specifically, HSP groups the cells to divide the WSI into a collection of sub-regions. For each group, we conduct intra-group information interaction to learn the cell spatial distribution. By aggregating features for each group and repeating the above process at higher levels, HSP hierarchical models cell clouds. The pseudocode of HSP is shown as Algorithm 1.

Given the coordinates $C = \{c^{(i)}\}_{i=1}^{N_{total}}$ and features $F_{cell} = \{f_{cell}^{(i)}\}_{i=1}^{N_{total}}$ of a collection of cells, group anchors $K = \{k^{(i)}\}_{i=1}^{N_k}$ are generated by Farthest Point Sampling (FPS) (Qi et al., 2017b), where $c^{(i)} \in \mathbb{R}^2$ and $f_{cell}^{(i)}$ are the coordinate and feature of the $i$-th cell, $N_k$ is the number of group anchors, and $k^{(i)} \in \mathbb{R}^2$ is the $i$-th anchor. To preserve the spatial distribution of cells as much as possible, we incorporate cell category information into the anchor generating. For any group anchor $k^{(i)}$, $2N_{total}/N_k$ nearest neighbors are assigned to it and marked as a group.

Cells assigned to the same group are spatially proximate, thus forming a local region. Consequently, we further update the cell features within each group and aggregate them to obtain features that describe the local cell spatial distribution. However, multiple clusters of spatially adjacent cells may be assigned to the same group. We introduce a semantic-spatial aware filter to address this issue and provide a detailed visualization and analysis in Section 5.

The semantic-spatial aware filter comprehensively considers the semantic similarity and spatial distance of cells within the same group. For each group, the coordinates of the group anchor and the mean feature of the cells within the group are used as references. Then, a similarity score is computed for each cell:

$$S_{sim}^{(i)} = \exp\left(-\|c^{(i)} - c_{ref}\|\right) \frac{(f_{cell}^{(i)})^T f_{ref}}{N_{dim}}, \tag{4}$$

where $S_{sim}^{(i)}$ is the similarity score of the $i$-th cell, $f_{ref}$ denotes the the mean feature, $c_{ref}$ denotes the coordinate of the group kernel, and $\|\cdot\|$ denotes the euclidean distance. We introduce a threshold $\lambda_{sim}$ to generate a filter $M$. If $S_{sim}^{(i)} < \lambda_{sim}$, $M^{(i)} = 0$ and the $i$-th cell is marked for discard.

For each cell, we calculate attention weights with respect to other cells within the same group to update its feature. The attention is implemented as vector attention (Zhao et al., 2021). Moreover, the positional relationships among cells are critical spatial information. Therefore, we incorporate relative coordinates into our calculations. Assuming that the $i$-th cell within the group $G$, the infor-

mation interaction between this cell and other cells is defined as follows:

$$S_{att}^{(i,j)} = M^{(j)} \mathcal{F}_{att}(\mathcal{W}_q f_{cell}^{(i)} - \mathcal{W}_k f_{cell}^{(j)} + E(c^{(i)} - c^{(j)})),$$
$$(f_{cell}^{(i)})' = \sum_G \delta(S_{att}^{(i,j)})(\mathcal{W}_v f_{cell}^{(j)} + E(c^{(i)} - c^{(j)})), \tag{5}$$

where $S_{att}^{(i,j)}$ is the attention vector, $\mathcal{F}_{att}$ is a Multilayer Perceptron (MLP), $(f_{cell}^{(i)})'$ is the updated feature of the $i$-th cell, $\delta$ denote the softmax and normalization for attention vectors, $\mathcal{W}_\cdot$ denotes a linear projection layer, and $E$ denotes a two layer MLP that maps the dimension of related distance to the same of features. Cells can perceive the spatial distribution by stacking layers as described in Equation 5.

HSP further introduces a hierarchical architecture to model the cell spatial distribution in a bottom-up manner. Specifically, HSP consists of multiple levels, each of which models the local spatial distribution of cells at a specific scale. Higher-level features are derived from lower-level features by mean aggregation and are subsequently re-grouped and undergo attention to model cell spatial distribution over a larger region. Finally, the feature of the WSI is the maximum aggregation of features at the last level.

**From CCFormer to FusedCCFormer: Fused with the appearance representation.** CCFormer is built on the cell cloud and produces a WSI-level representation of cell spatial distribution. Notably, employing a foundation model to extract patch-level features can effectively construct a WSI-level appearance representation, which has been shown to enhance performance in downstream tasks such as survival prediction (Chen et al., 2024). In this work, we explore fusing CCFormer with patch embeddings from the foundation model, termed **FusedCCFormer**, to achieve a more comprehensive and expressive WSI representation. Specifically, patch embeddings from the foundation model are global-mean pooled across the slide to get the WSI appearance feature $f_{app}^{WSI}$. The final WSI feature $f_{WSI}$ is defined as $f_{cell}^{(WSI)} + \beta \cdot E(f_{app}^{(WSI)})$, where $\beta$ is the weight of the appearance term.

## 5 EXPERIMENTS

### 5.1 EXPERIMENT SETTINGS

**Datasets & Metrics.** We conduct extensive experiments of survival prediction and cancer staging on multiple cancer datasets of TCGA. In our experiments, we follow the usual practice to evaluate the performance of survival prediction (Chen et al., 2021; Nakhli et al., 2023a) and cancer staging (Li et al., 2024; Chan et al., 2023) by C-Index and Macro-F1 with 5-fold cross-validation, respectively. In particular, we report the results of 5-fold cross-validation for each experiment. All reported results in percentages refer to relative improvements.

**Baselines.** Global mean pooling (MeanPool) and global max pooling (MaxPool) are employed as baselines. In addition, we compare CCFormer with SOTA WSI analysis methods. For MIL-based methods, we compare with TransMIL (Shao et al., 2021), ABMIL (Ilse et al., 2018), CLAM (Lu et al., 2021), DSMIL (Li et al., 2021), $R^2$T-MIL (Tang et al., 2024), HDMIL (Dong et al., 2025), and MambaMIL (Yang et al., 2024). As MIL methods are highly influenced by the pre-trained feature extractor, we utilize the pre-trained UNI (Chen et al., 2024), a SOTA self-supervised model for pathology, to extract patch features. For graph-based methods, we compare with Patch-GCN (Chen et al., 2021), WiKG (Li et al., 2024), and PAMOE (Wu et al., 2025). Besides, we also compare with Pixel-Mamba (Qiu et al., 2025), which is an SOTA end-to-end WSI representation method. In addition, models for learning point clouds can be adaptively applied to cell clouds. Thus, we compare CCFormer with SOTA point cloud methods, including PointNet (Qi et al., 2017a), PointNet++ (Qi et al., 2017b), and Point Transformer v3 (PTv3) (Wu et al., 2024). The baselines are implemented using their released code and the same split files, unless otherwise specified.

**Implementation Details.** Please refer to the Appendix.

### 5.2 MAIN RESULTS

Table 1 and Figure 5 report the results of survival prediction and cancer staging, respectively. Methods based on cell clouds achieve competitive results with MIL-based, graph-based, and end-to-end

Table 1: **Comparison of survival prediction in C-Index** (↑). CCFormer outperforms other SOTA methods. The best results are highlighted in bold, and the second-best results are in underlined.

| | Method | Params. | BLCA | BRCA | COAD READ | LUAD | PAAD | STAD |
|---|---|---|---|---|---|---|---|---|
| **Patch feature MIL** | MeanPool (Patch) | 4.1K | $0.610 \pm 0.024$ | $0.643 \pm 0.031$ | $0.629 \pm 0.186$ | $0.571 \pm 0.057$ | $0.672 \pm 0.097$ | $0.579 \pm 0.085$ |
| | MaxPool (Patch) | 4.1K | $0.510 \pm 0.038$ | $0.589 \pm 0.063$ | $0.585 \pm 0.078$ | $0.480 \pm 0.037$ | $0.404 \pm 0.122$ | $0.474 \pm 0.102$ |
| | TransMIL (Shao et al., 2021) | 2.7M | $0.600 \pm 0.061$ | $0.663 \pm 0.058$ | $0.634 \pm 0.105$ | $0.587 \pm 0.076$ | $0.636 \pm 0.121$ | $0.581 \pm 0.081$ |
| | ABMIL (Ilse et al., 2018) | 0.9M | $0.609 \pm 0.028$ | $0.656 \pm 0.055$ | $0.668 \pm 0.167$ | $0.614 \pm 0.066$ | $0.696 \pm 0.080$ | $0.650 \pm 0.098$ |
| | CLAM (Lu et al., 2021) | 0.8M | $0.617 \pm 0.024$ | $0.656 \pm 0.104$ | $0.665 \pm 0.100$ | $0.619 \pm 0.112$ | $0.697 \pm 0.084$ | $0.641 \pm 0.118$ |
| | DSMIL (Li et al., 2021) | 0.2M | $0.601 \pm 0.045$ | $0.612 \pm 0.070$ | $0.647 \pm 0.035$ | $0.598 \pm 0.117$ | $0.677 \pm 0.105$ | $0.615 \pm 0.078$ |
| | HDMIL (Dong et al., 2025) | 0.9M | $0.597 \pm 0.070$ | $0.605 \pm 0.086$ | $0.607 \pm 0.029$ | $0.620 \pm 0.092$ | $\underline{0.748 \pm 0.075}$ | $0.621 \pm 0.108$ |
| | MambaMIL (Yang et al., 2024) | 4.2M | $0.632 \pm 0.048$ | $0.659 \pm 0.069$ | $0.672 \pm 0.066$ | $0.631 \pm 0.139$ | $0.701 \pm 0.082$ | $0.628 \pm 0.087$ |
| | $R^2$T-MIL (Tang et al., 2024) | 2.7M | $0.648 \pm 0.031$ | $0.640 \pm 0.103$ | $0.697 \pm 0.051$ | $0.655 \pm 0.101$ | $0.712 \pm 0.072$ | $0.638 \pm 0.117$ |
| **Graph** | Patch-GCN (Chen et al., 2021) | 1.4M | $0.597 \pm 0.022$ | $0.628 \pm 0.036$ | $0.634 \pm 0.121$ | $0.617 \pm 0.043$ | $0.668 \pm 0.115$ | $0.563 \pm 0.048$ |
| | WiKG (Li et al., 2024) | 2.0M | $0.638 \pm 0.030$ | $0.649 \pm 0.036$ | $0.722 \pm 0.069$ | $0.632 \pm 0.038$ | $0.661 \pm 0.112$ | $0.672 \pm 0.089$ |
| | PAMOE+Patch-GCN (Wu et al., 2025) | 15.5M | $0.593 \pm 0.028$ | $0.589 \pm 0.026$ | $0.643 \pm 0.086$ | $0.620 \pm 0.049$ | $0.640 \pm 0.079$ | $0.549 \pm 0.046$ |
| **Point Cloud** | MeanPool (Cell) | 1.5K | $0.535 \pm 0.045$ | $0.573 \pm 0.070$ | $0.639 \pm 0.063$ | $0.552 \pm 0.068$ | $0.647 \pm 0.045$ | $0.569 \pm 0.074$ |
| | MaxPool (Cell) | 1.5K | $0.476 \pm 0.024$ | $0.571 \pm 0.055$ | $0.514 \pm 0.156$ | $0.535 \pm 0.035$ | $0.544 \pm 0.065$ | $0.527 \pm 0.090$ |
| | PointNet (Qi et al., 2017a) | 3.5M | $0.633 \pm 0.025$ | $0.665 \pm 0.021$ | $0.732 \pm 0.044$ | $0.638 \pm 0.012$ | $0.715 \pm 0.047$ | $0.682 \pm 0.075$ |
| | PointNet++ (Qi et al., 2017b) | 1.5M | $0.613 \pm 0.038$ | $0.656 \pm 0.036$ | $0.743 \pm 0.028$ | $0.645 \pm 0.020$ | $0.702 \pm 0.043$ | $0.633 \pm 0.055$ |
| | PTv3 (Wu et al., 2024) | 39M | $0.553 \pm 0.039$ | $0.536 \pm 0.054$ | $0.616 \pm 0.069$ | $0.591 \pm 0.050$ | $0.631 \pm 0.114$ | $0.560 \pm 0.023$ |
| | Pixel-Mamba (Qiu et al., 2025)[†] | 6.2M | $\underline{0.651} \pm 0.049$ | $0.671 \pm 0.073$ | - | $0.647 \pm 0.033$ | - | - |
| | **CCFormer (ours)** | 2.0M | $0.641 \pm 0.019$ | $\underline{0.696 \pm 0.056}$ | $\underline{0.782 \pm 0.071}$ | $\underline{0.661 \pm 0.034}$ | $\underline{0.748 \pm 0.056}$ | $\underline{0.695 \pm 0.076}$ |
| | **FusedCCFormer (ours)** | 2.7M | $\mathbf{0.667 \pm 0.023}$ | $\mathbf{0.729 \pm 0.066}$ | $\mathbf{0.795 \pm 0.068}$ | $\mathbf{0.672 \pm 0.045}$ | $\mathbf{0.771 \pm 0.072}$ | $\mathbf{0.702 \pm 0.063}$ |

[†]Results are directly taken from the original paper Qiu et al. (2025).

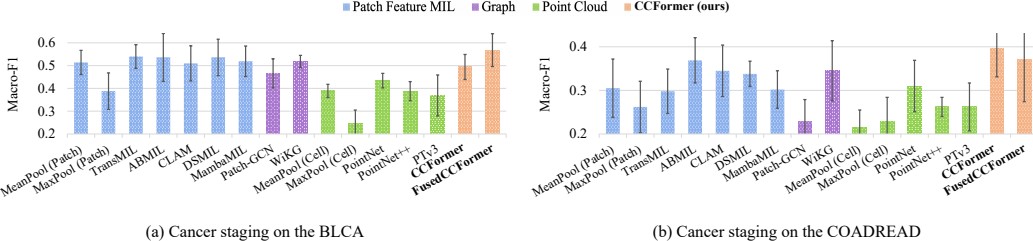

(a) Cancer staging on the BLCA        (b) Cancer staging on the COADREAD

Figure 5: **Comparison of cancer staging with SOTA methods on (a) BLCA and (b) COAD-READ in Macro-F1** (↑). FusedCCFormer outperforms SOTA methods on both cancers.

methods. Results indicate that patient survival risk and cancer stages are highly related to cell spatial distribution, which aids in the analysis of WSIs and further enhances the accuracy of downstream tasks. Moreover, by fusing cell spatial distribution representations with appearance representations, FusedCCFormer significantly outperforms existing SOTA methods.

**Survival Prediction.** Table 1 reports survival prediction results across multiple cancers. Compared to MIL-based and graph-based SOTA methods, CCFormer achieves the C-Index with improvements of 1% to 8% on the BLCA, BRCA, COADREAD, LUAD, PAAD, and STAD. Compared with end-to-end methods, CCFormer significantly outperforms Pixel-Mamba across most cancer types, except on BLCA, where it is lower by 0.01. FusedCCFormer further delivers substantial C-index improvements across all cancer types. Specifically, compared to SOTA baselines, FusedCCFormer achieves the C-Index with improvements of 2% to 10% on multiple cancers.

**Cancer Staging.** Figure 5 (a) reports the results of cancer staging on the BLCA. CCFormer outperforms PointNet, PointNet++, and PTv3 with 14%, 27%, and 34%, respectively. The experiments also show that representing WSIs by cell spatial distribution alone is insufficient to fully capture the complexity of BLCA staging. Although patch-based methods achieve substantially higher Macro-F1 than methods based on cell cloud alone, FusedCCFormer achieves higher Micro-F1 and surpasses the best patch-based method by 5%. This indicates that cell spatial distribution representations encode staging-relevant features that are complementary to WSI appearance features.

Table 2: **The number of detected cells for each cancer dataset.** HLLR can effectively increase the number of detected cells, further improving the performance of CCFormer in downstream tasks.

| HLLR | BLCA | BRCA | COAD READ | LUAD | PAAD | STAD |
|------|------|------|-----------|------|------|------|
| ✗ | 449M | 623M | 396M | 481M | 213M | 692M |
| ✓ | 475M | 663M | 424M | 510M | 226M | 736M |

Table 3: **Ablation Studies of NIE and spatial-semantic aware filtering on BRCA in C-Index (↑).** **Left**: Neighboring Information Embedding (NIE) significantly improves performance. **Mid**: Hierarchical Spatial Perception (HSP) ablation demonstrates benefits of spatial-semantic aware filtering. **Right**: CCFormer exhibits excellent performance across a broad range of $N_k$ and $N_{basic}$. An appropriate $N_{basic}$ further improves the modeling of local cell spatial distributions, enabling CCFormer to significantly outperform other SOTA methods.

| Coordinate | Type | NIE | C-Index (↑) |
|------------|------|-----|-------------|
| ✗ | ✓ | ✗ | $0.573 \pm 0.070$ |
| ✓ | ✓ | ✗ | $0.665 \pm 0.039$ |
| ✓ | ✗ | ✓ | $0.678 \pm 0.021$ |
| ✓ | ✓ | ✓ | $\mathbf{0.696} \pm 0.056$ |

| Filter Type | $\lambda_{sim}$ | C-Index (↑) |
|-------------|-----------------|-------------|
| None | - | $0.668 \pm 0.043$ |
| Semantic | 0.5 | $0.691 \pm 0.044$ |
| Spatial | 0.5 | $0.655 \pm 0.040$ |
| Spatial + Semantic | 0.9 | $0.652 \pm 0.043$ |
| Spatial + Semantic | 0.5 | $\mathbf{0.696} \pm 0.056$ |

| $N_k$ | $N_{basic}$ | C-Index (↑) |
|-------|-------------|-------------|
| 2048 | 4 | $0.652 \pm 0.052$ |
| 2048 | 16 | $\mathbf{0.696} \pm 0.056$ |
| 2048 | 32 | $0.662 \pm 0.023$ |
| 1024 | 16 | $0.683 \pm 0.045$ |
| 4096 | 16 | $0.663 \pm 0.059$ |

Figure 5 (b) reports the results of cancer staging on the COADREAD. CCFormer outperforms Point-Net, PointNet++, and PTv3 with 28%, 51%, and 51%, respectively. Compared to SOTA MIL-based methods, including TransMIL, ABMIL, CLAM, DSMIL, and MambaMIL, CCFormer achieves significant improvement with 33%, 7%, 15%, 17%, and 31% respectively. In addition, compared to SOTA graph-based methods, including Patch-GCN and WiKG, CCFormer achieves improvement with 73% and 15%, respectively. The experiments also show that the global mean pooling can not provide effective WSI features for cancer staging on the COADREAD, leading to a slight decrease in Macro-F1 after combining CCFormer with MeanPool.

## 5.3 ABLATION STUDIES

**Human-in-the-Loop Label Refinement.** To comprehensively assess the impact of HLLR on the representation of cell spatial distribution, we evaluate with two metrics: 1) the number of detected cells, which shows the effect of HLLR on cell cloud construction, and 2) survival prediction performance, which directly evaluates its influence on downstream clinical endpoints. As shown in Table 2, HLLR substantially increases the number of detected cells. Specifically, across BLCA, BRCA, COADREAD, LUAD, PAAD, and STAD, after finetuning the cell detection and classification model with HLLR, the model yields an approximately 6% increase in detected cells. Since each cancer dataset contains 200 to 700 million cells, tens of millions of additional cells per cancer type are detected, markedly strengthening the representation of cell spatial distribution. Moreover, HLLR improves survival prediction by an average of 1.2% in C-Index across the six cancer datasets. For FusedCCFormer, the effect is more pronounced, with an average C-index gain of 2.4%.

**Neighboring Information Embedding.** Table 3 (Left) reports the results of ablation studies on NIE. NIE improves the C-Index on BRCA from 0.665 to 0.696. The ablation studies further demonstrate that cell coordinates and cell types are crucial for high-performance representations of cell spatial distribution. We cluster cells based on $F_{ld}$ and $F_{gd}$ via K-Means to visualize NIE. As shown in Figure 6 (a), cells with different neighboring cell spatial distributions are distinguished. Specifically, cancer cell clusters, immune cell clusters, mixed cell regions, and other cell clusters are identified.

**Hierarchical Spatial Perception.** Table 3 (Mid) evaluates the semantic–spatial-aware filter. By forming more coherent local groups than the baseline, it increases the BRCA C-index from 0.668 to 0.696. In addition, the filter with a large threshold removes lots of important cells, resulting in a decrease in C-Index. Table 3 (Right) analyzes the influence of $N_k$ and $N_{basic}$. CCFormer maintains robust performance across various settings of $N_k$ and $N_{basic}$, except in extreme values ($N_{basic} = 4$). An appropriate $N_{basic}$ further improves the modeling of local cell spatial distributions. Notably, all

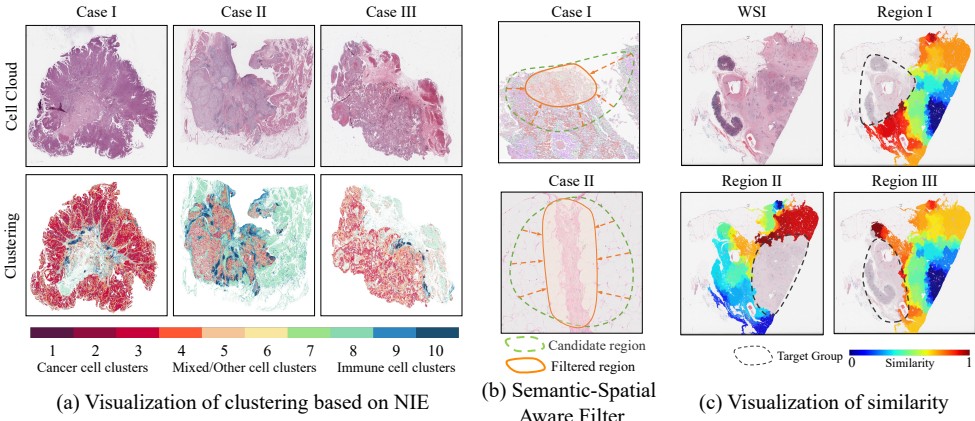

Figure 6: **Visualization.** (a) Visualization of clustering based on NIE. Cancer cell clusters, immune cell clusters, mixed cell clusters, and other cell clusters are identified. (b) Visualization of the semantic-spatial aware filter. The filter drives the model to attend to important subregions within each candidate region, thereby enhancing local representations of cell spatial distribution. (c) Visualization of similarity. CCFormer effectively learns the semantics of different regions.

experiments in the main results employ the default $N_{basic}$, verifying the strong parameter robustness of CCFormer.

Figure 6 (b) shows the visualization of the semantic-spatial aware filter. Automatically generated candidate regions can inadvertently merge spatially discontinuous or semantically heterogeneous subregions into a single group, obscuring representation of local cell spatial distribution. The semantic-spatial aware filter identifies the subset most similar to the group anchor within each candidate region, thereby improving the performance of CCFormer.

**Visualization of Similarity.** To illustrate how CCFormer understands relationships across different regions, we visualize the similarity among groups at the last level. Specifically, we select a group and visualize the similarity between this group and others. In addition, the attention scores are mapped back to the input cell cloud. As shown in Figure 6 (c), CCFormer comprehends the semantic relationships among regions. Local tissue regions with similar structures exhibit high similarity.

## 6 CONCLUSION

In this paper, we claim the importance of modeling cell distribution in solving pathology downstream tasks. Specifically, we argue that the collection of cells within a WSI can be regarded as a cell cloud and propose a new scheme consisting of cell detection and cell cloud modeling. First, we propose the Human-in-the-Loop Label Refinement (HLLR) to finetune the pretrained cell detection and classification model. Then, a novel hierarchical cell cloud Transformer (CCFormer) is proposed to model the spatial distribution of cells. Specifically, we propose a novel Neighboring Information Embedding (NIE) to embed the neighborhood cell distribution at the cell level and a novel Hierarchical Spatial Perception (HSP) to progressively perceive and aggregate cell spatial distribution information. The clinical analysis validates that cell clouds can be used to construct effective clinical indicators. In addition, extensive experiments verify that cell cloud is an effective representation of slide and CCFormer outperforms other SOTA methods. This work provides a new insight for WSI analysis from the cell cloud perspective, marking a significant milestone in the advancement of computational pathology.

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

# A    THE USE OF LARGE LANGUAGE MODELS

All core technical contributions in this work are conceived and developed exclusively by the authors without assistance from large language models (LLMs). LLMs are used solely for manuscript polishing. All LLM-polished technical descriptions and result analyses are rigorously reviewed and verified by the authors.

# B    IMPLEMENTATION

## B.1    CELL DETECTION AND CLASSIFICATION

**Finetuning with Human-in-the-Loop Label Refinement.** More than 36k credible patches (3 million cells) are selected to build the fine-tuning dataset. In addition, 743 samples (141,649 cells) are labeled by experts to further enhance the performance of domain-adaptation finetuning.

**Preprocessing of WSIs.** We pre-process and detect cells with the same workflow for each WSI:

- **Region of Interest.** We employ CLAM (Lu et al., 2021) to extract Regions of Interest (ROI) in WSIs to reduce the number of pixels that need to be processed for cell detection and classification. WSIs are fixed at 40x magnification to ensure that each cell has sufficient detail. Due to the limitations of memory, the ROI of each WSI is divided into patches of $512 \times 512$ pixels for further processing.

- **Cell Detection and Classification.** The data distribution of PanNuke (Gamper et al., 2020) partially overlaps with that of TCGA. More importantly, PanNuke covers 19 different organs, enabling the training of a cell detection and classification model that generalizes well across a wide range of tissue types, which is essential for supporting downstream tasks on multiple cancers. Therefore, we pre-train DPA-P2PNet (Shui et al., 2024) on PanNuke to obtain a more accurate and broadly applicable cell detection and classification model. In this work, we focus on analyzing the spatial relationship among neoplastic cells, inflammatory cells, and other cells. Therefore, we integrate the cell types of PanNuke in the pre-training. Specifically, cells with the type of connective, dead, and epithelial in PanNuke are labeled as others.

- **Merging Patches.** We further merge the prediction on patches and generate results for each WSI. Since the same cell might be split into multiple patches and predicted repeatedly by the model, we merge cells that are close to the patch boundaries. Specifically, the image resolution of WSI at 40x magnification is about 0.25 $\mu$m per pixel and the cell diameter is approximately 10 $\mu$m. Therefore, we select cells that are less than 24 pixels away from the patch boundaries and merge cells of the same type that are less than 12 pixels apart.

## B.2    CLINICAL ANALYSIS

We introduce a hyper-parameter vector $\alpha$ to control the weight of each component in the Cell Proportion Score (CPS). Additionally, each component of CPS is subjected to min-max normalization. Table 4 reports log-rank test p-values for COADREAD and PAAD with different settings of $\alpha$. The results show that each metric, considered individually, successfully stratifies patients into low- and high-risk groups. Furthermore, for COADREAD, combining components ($\alpha = [0.50, 0.25, 0.25]^T$) further reduces the log-rank p-values, indicating stronger discrimination of patients' survival risk.

Table 4: **Clinical analysis in log-rank p-values** ($\downarrow$). Across a wide range of $\alpha$, clinical indicators derived from cell statistics robustly stratify patients into low- and high-risk groups.

| $\alpha$ | COADREAD | PAAD |
|---|---|---|
| $[0, 0, 1]^T$ | 1.42e-02 | 1.38e-02 |
| $[0, 1, 0]^T$ | 4.59e-02 | 9.91e-03 |
| $[1, 0, 0]^T$ | 3.65e-03 | **1.43e-03** |
| $[0.50, 0.25, 0.25]^T$ | **1.54e-03** | 3.87e-03 |

### B.3 CCFORMER

The scale factor $\lambda_r = 4$ and the discrete number $N_d = 3$ in NIE. For HSP, we set the number of perception levels $L = 3$, the initial number of group anchors $N_k = 2048$, and the basic number of points within each group $N_{basic} = 16$. In each level, the input features are updated twice based on Equation 5 of the main text, and the input dimension is expanded to twice its original size in the group-wise aggregation. The semantic-spatial aware filter threshold $\lambda_{sim}$ is set to 0.5 by default. Due to cancer-specific heterogeneity in cell spatial distributions, we slightly adjust $\lambda_{sim}$ per cancer type around this default.

### B.4 TRAINING

We follow the splitting method of SurvPath (Jaume et al., 2024) that WSIs are split into 5 folds according to the sample sit. Negative log-likelihood survival loss and cross-entropy loss are employed for training models on survival prediction and cancer staging, respectively.

Baselines (Ilse et al., 2018; Shao et al., 2021; Chen et al., 2021; Li et al., 2024; Qi et al., 2017a;b; Wu et al., 2024; Yang et al., 2024; Lu et al., 2021; Li et al., 2021) are implemented with their released codes. MIL-based methods are optimized using RAdam (Liu et al., 2019), a batch size of 1, a learning rate of $2 \times 10^{-4}$, $1 \times 10^{-3}$ weight decay, and epochs of 20. Patch-GCN (Chen et al., 2021) and WiKG (Li et al., 2024) are optimized using their default hyper-parameters. Point cloud methods are optimized using Adam (Kingma, 2014), a batch size of 8, a learning rate of $1 \times 10^{-3}$, cosine annealing learning rate decay to $1 \times 10^{-4}$, and epochs of 150. For CCFormer, we adjust the learning rate, the semantic-spatial aware filter threshold $\lambda_{sim}$, and the dropout ratio for each cancer based on the same training parameters as the point cloud methods. Due to the significant difference in convergence speed between cell cloud methods and MIL-based methods, the combination of CCFormer and MeanPool (Patch) fails to model the cell spatial distribution if model parameters are randomly initialized. Therefore, the combination model loads the pre-trained CCFormer and freezes it during training. Only the two fully connected layers added for MeanPool (Patch) are optimized using Adam, a learning rate of $5 \times 10^{-4}$, cosine annealing learning rate decay to $1 \times 10^{-4}$, and epochs of 10.

Experiments are conducted on a single NVIDIA H20. The average cell identification time for a single WSI is 2.1 minutes. Following the preprocessing method of large-scale point clouds, we perform grid sampling on cell clouds. Further sampling is performed for mini-batch training. The average floating-point operations are 21.3 GFlops for each WSI during training.

## C ADDITIONAL DISCUSSION AND EXPERIMENTS

**Additional Discussion of Cell-Level Methods.** Notably, prior studies have tried to represent and analyze whole-slide images (WSIs) from a cell-level perspective, such as Co-Pilot (Nakhli et al., 2023b) and Ceograph (Wang et al., 2023). These methods typically use graphs to model cell relationships. However, due to the substantial computational overhead, these graph-based approaches are limited to patch-level analysis. For example, Ceograph samples 1024x1024 patches from ROIs and builds a graph for each patch. In contrast, CCFormer directly analyzes the entire WSI as a unified cell cloud, enabling holistic representation learning and fully capturing cell-level relationships without patch-based constraints.

**Comparison with Hierarchical Patch-Based Methods.** In this work, we propose HSP to model cell spatial distribution in a bottom-up manner. The hierarchical structure has also been explored in patch-based methods. We compare CCFormer with HIPT (Chen et al., 2022) on the BRCA survival prediction. HIPT and CCFormer achieve C-Index of $0.614 \pm 0.045$ and $0.696 \pm 0.056$, respectively, indicating that hierarchical learning of cell spatial distributions retains a performance advantage.

**Limitations.** This work is designed to be applied to as many types of cancer as possible. Therefore, we categorize cells into three basic types: neoplastic, inflammatory, and other. Experiments on survival prediction and cancer staging validate the rationality of this design.

This design also limits the performance of methods based on cell cloud in some cancers, which require fine-grained cell classification. Detailed results of cancer staging are shown in Table 5. For

Table 5: Comparison of cancer staging with SOTA methods on BLCA and COADREAD in ACC (↑), AUC (↑), and Macro-F1 (↑). The combination of CCFormer and the global mean pooling outperforms baselines on both cancers.

| Method | BLCA | | | COADREAD | | |
|---|---|---|---|---|---|---|
| | ACC | AUC | Macro-F1 | ACC | AUC | Macro-F1 |
| **Patch feature MIL** | | | | | | |
| MeanPool (Patch) | 0.482 ± 0.057 | 0.701 ± 0.040 | 0.514 ± 0.053 | 0.390 ± 0.044 | 0.561 ± 0.090 | 0.305 ± 0.067 |
| MaxPool (Patch) | 0.380 ± 0.072 | 0.662 ± 0.042 | 0.388 ± 0.080 | 0.351 ± 0.043 | 0.539 ± 0.084 | 0.262 ± 0.059 |
| TransMIL (Shao et al., 2021) | 0.491 ± 0.078 | 0.709 ± 0.059 | 0.540 ± 0.052 | 0.376 ± 0.063 | 0.564 ± 0.041 | 0.298 ± 0.051 |
| ABMIL (Ilse et al., 2018) | 0.515 ± 0.084 | **0.722** ± 0.070 | 0.536 ± 0.105 | 0.420 ± 0.063 | 0.583 ± 0.042 | 0.369 ± 0.052 |
| CLAM (Lu et al., 2021) | 0.512 ± 0.078 | 0.689 ± 0.048 | 0.510 ± 0.077 | 0.391 ± 0.055 | 0.579 ± 0.072 | 0.345 ± 0.059 |
| DSMIL (Li et al., 2021) | 0.550 ± 0.079 | 0.688 ± 0.066 | 0.536 ± 0.080 | 0.386 ± 0.025 | 0.560 ± 0.045 | 0.338 ± 0.029 |
| MambaMIL (Yang et al., 2024) | 0.534 ± 0.060 | 0.683 ± 0.051 | 0.519 ± 0.067 | 0.345 ± 0.049 | 0.556 ± 0.042 | 0.302 ± 0.043 |
| **Graph** | | | | | | |
| Patch-GCN (Chen et al., 2021) | 0.493 ± 0.053 | 0.644 ± 0.067 | 0.466 ± 0.064 | 0.319 ± 0.055 | 0.502 ± 0.073 | 0.229 ± 0.050 |
| WiKG (Li et al., 2024) | 0.523 ± 0.028 | 0.674 ± 0.033 | 0.518 ± 0.027 | 0.392 ± 0.047 | 0.563 ± 0.065 | 0.345 ± 0.069 |
| **Point Cloud** | | | | | | |
| MeanPool (Cell) | 0.420 ± 0.038 | 0.582 ± 0.046 | 0.389 ± 0.029 | 0.378 ± 0.084 | 0.475 ± 0.040 | 0.216 ± 0.039 |
| MaxPool (Cell) | 0.366 ± 0.025 | 0.521 ± 0.012 | 0.246 ± 0.059 | 0.404 ± 0.092 | 0.547 ± 0.072 | 0.229 ± 0.055 |
| PointNet (Qi et al., 2017a) | 0.436 ± 0.027 | 0.560 ± 0.023 | 0.434 ± 0.032 | 0.430 ± 0.081 | 0.561 ± 0.055 | 0.310 ± 0.059 |
| PointNet++ (Qi et al., 2017b) | 0.436 ± 0.048 | 0.555 ± 0.067 | 0.388 ± 0.042 | 0.397 ± 0.060 | 0.540 ± 0.057 | 0.293 ± 0.061 |
| PTv3 (Wu et al., 2024) | 0.420 ± 0.075 | 0.606 ± 0.085 | 0.369 ± 0.090 | **0.443** ± 0.062 | 0.514 ± 0.056 | 0.262 ± 0.055 |
| **CCFormer (ours)** | 0.501 ± 0.057 | 0.635 ± 0.064 | 0.494 ± 0.055 | 0.429 ± 0.031 | 0.593 ± 0.057 | **0.396** ± 0.065 |
| **FusedCCFormer (ours)** | **0.569** ± 0.072 | 0.702 ± 0.069 | **0.568** ± 0.072 | 0.420 ± 0.063 | **0.595** ± 0.061 | 0.371 ± 0.097 |

Table 6: Cancer abbreviation and full name cross-reference table.

| Abbreviation | Full Name | Abbreviation | Full Name |
|---|---|---|---|
| BLCA | Bladder Urothelial Carcinoma | PAAD | Pancreatic Adenocarcinoma |
| BRCA | Breast Invasive Carcinoma | READ | Rectum Adenocarcinoma |
| COAD | Colon Adenocarcinoma | STAD | Stomach Adenocarcinoma |
| LUAD | Lung Adenocarcinoma | | |

KIRC, the nuclear grade is significantly associated with patient survival risk (Frank et al., 2002; Sorbellini et al., 2005). For Lower Grade Gliomas (LGG), mitotic figures, nuclear atypia, necrosis, and microvascular proliferation are important factors in diagnosing the survival risk of patients. (Duregon et al., 2016) For BLCA, depth of invasion is a critical indicator in cancer staging (Dyrskjøt et al., 2023). The stage of BLCA can be effectively judged by further categorizing neoplastic cells based on whether there is muscle invasion. Therefore, further subclassification of cells represents an effective strategy to enhance the performance of methods based on cell cloud.

## D  SYMBOL EXPLANATION

Table 6 summarizes the cross-reference of cancer abbreviations and full names.

## E  COMPARISON OF COMPUTATIONAL EFFICIENCY

**Preprocessing.** The comparison between UNI and DPA-P2PNet is summarized in the Table 7. For UNI, patches are resized to 224, which is the default input size. For DPA-P2PNet, the default input size is 512, as detailed in Appendix B. The WSI-level results are averaged over WSIs.

Compared with DPA-P2PNet, UNI has 5.5 times more parameters, which substantially increases the computational cost of MIL-based methods in the preprocessing stage. Although the default input

Table 7: **Comparison of preprocessing computational efficiency in throughput (↑), FLOPs (↓), and runtime (↓).** MIL-based methods rely on foundation models such as UNI to extract patch features, while CCFormer depends on cell detection and classification models like DPA-P2PNet to infer WSIs and construct cell clouds. Since DPA-P2PNet is more lightweight than FMs, the preprocessing stage of CCFormer has comparable or even lower total floating-point operations and faster processing speed compared with MIL-based methods.

| Method | Parameters | Patch Size | Throughput (patches/s) | FLOPs (per patch) | Runtime (per WSI, s) | FLOPs (per WSI) |
|---|---|---|---|---|---|---|
| UNI (Chen et al., 2024) | 303M | 224 | 278 | 60 GFLOPs | 41 | 672 TFLOPs |
| UNI V2 (Chen et al., 2024) | 681M | 224 | 114 | 180 GFLOPs | 99 | 2.0 PFLOPs |
| DPA-P2PNet (Shui et al., 2024) | 55M | 512 | 160 | 87 GFLOPs | 70 | 981 TFLOPs |
| DPA-P2PNet (Shui et al., 2024) | 55M | 256 | 612 | 22 GFLOPs | 18 | 245 TFLOPs |

Table 8: **Comparison of computational efficiency of WSI-level tasks in runtime (↓) and FLOPs (↓).**

| Method | Memory (GB) | Runtime (per WSI, s) | FLOPs (per WSI) |
|---|---|---|---|
| ABMIL (Ilse et al., 2018) | 2.4 | 0.0012 | 9 GFLOPs |
| TransMIL (Shao et al., 2021) | 9.3 | 0.0116 | 63 GFLOPs |
| PointNet (Qi et al., 2017a) | 2.7 | 0.0201 | 17 GFLOPs |
| CCFormer (ours) | 67 | 0.2050 | 117 GFLOPs |

patch size of DPA-P2PNet is 2.3 times that of UNI, the inference cost per patch remains comparable between UNI and DPA-P2PNet.

In addition, the input size for cell detection and classification can be reduced from 512 at 40× magnification to 256 at 20× magnification to further improve computational efficiency. Our computational analysis shows that, under this setting, the average WSI inference time based on DPA-P2PNet is only 0.44× that of UNI. To assess the impact of this change on cell detection performance at 40× and 20×, we conduct an analysis of cell detection performance on the Lizard dataset. Specifically, DPA-P2PNet achieves F1 scores of 88.08 and 87.90 at 40× and 20× magnification, respectively. This shows that performing cell inference at a lower magnification can substantially accelerate cell cloud inference without compromising accuracy. In contrast, more advanced foundation models further increase the number of parameters, imposing a substantial computational burden on MIL-based methods. For example, UNI V2 contains 681M parameters. The inference cost of UNI V2 for a single WSI increases to 2.0 PFLOPs.

**WSI-level tasks.** We compare the inference computational cost of MIL-based methods and cell cloud-based methods on WSI-level tasks in Table 8. The runtime is measured as the average forward-pass time with a batch size of 1. Both runtime and FLOPs exclude preprocessing and data loading.

PointNet exhibits memory usage comparable to MIL-based methods. The FLOPs per WSI of PointNet are also significantly lower than those of TransMIL. Moreover, PointNet achieves substantially better performance than both ABMIL and TransMIL as shown in Table 1. These results indicate that representing WSIs as cell clouds is an effective approach that does not incur higher computational costs than MIL-based methods.

Although CCFormer requires more GPU memory, longer runtime, and higher FLOPs than MIL-based methods that only perform feature aggregation, it can still infer WSIs within one second on a single GPU with at least 80 GB of memory. Moreover, CCFormer offers several advantages: 1) it explicitly models the cell spatial distribution of WSIs rather than merely aggregating patch features, and 2) it achieves significantly better survival prediction performance across multiple cancer types compared to MIL-based methods.

