# OpenReview forum: "From Histopathology Images to Cell Clouds: Learning Slide Representations with Hierarchical Cell Transformer"
_ICLR.cc/2026/Conference — Submitted to ICLR 2026_

### Official Review · Reviewer_aWo2 · 2025-10-22

**Soundness:** 3
**Presentation:** 3
**Contribution:** 3
**Rating:** 4
**Confidence:** 4

**Summary:**

This work presents a new method, named Hierarchical Cell Cloud Transformer, to address the WSI-level tasks, including survival analysis and cancer staging. The authors designed the Human-in-the-Loop Label Refinement strategy to reduce the cost of manual annotation for building the cell segmentation model. Besides, they introduce a novel Neighboring Information Embedding (NIE) to capture neighborhood cell distribution at the cell level, and a Hierarchical Spatial Perception (HSP) method to model cell spatial distribution information in a bottom-up manner. Clinical analysis and extensive experiments on multiple public WSI datasets were performed to demonstrate the efficacy of the cell cloud framework and the CCFormer model.

**Strengths:**

The manuscript presents a new idea for whole slide image analysis from a single-cell perspective. The proposed method captures the spatial information of cells, such as cell interaction, by the hierarchical Cell Cloud Transformer, which can enhance the WSI-level representation. The proposed method outperforms existing methods significantly across multiple datasets and tasks. This work might provide a new insight into WSI analysis for the research community.

**Weaknesses:**

The proposed approach might be complex, which involves cell segmentation, patch classification, feature representation and aggregation, and WSI prediction. It is not end-to-end and might be computationally extensive. Experimental analysis might not be very comprehensive, such as missing results and evaluation metrics.

**Questions:**

1)	My major concern about this work is that the proposed method proceeds in two steps: first cell segmentation, and then WSI-level analysis. As mentioned by the authors, each cancer dataset contains 200 to 700 million cells. There is inevitably a significant computational cost for segmentation, which largely hinders real-world applications. Please elaborate on the test time per WSI in the manuscript.
2)	In the hierarchical spatial perception, the authors applied mean aggregation at the low levels, but used maximum aggregation at the last level. Besides, for the appearance feature, the authors applied global-average pooling, which is also different from the last-level operation. Please elaborate on this.
3)	Some results are missing in Table 1 without any explanation, such as PAMOE. As PAMOE is publicly available, it is possible to run such experiments.
4)	Did the authors borrow results from existing works? The results of MambaMIL in the published paper are not the same as those in Table 1, but the results of PAMOE are the same as those in Table 1. I am quite fused. Moreover, PAMOE used LGG dataset for evaluation and achieved promising results (i.e., 0.793) for survival prediction, however this work skips the comparison.
5)	For evaluation metrics for cancer staging, the authors used the Macro-F1 score only. However, accuracy and AUC were commonly adopted in previous works. Please supplement these as well. Please specify whether the improvement in the percentage format is in absolute or relative terms.

---

> ### Author Response · Authors · 2025-11-25
> **Key issue: the computational efficiency comparison [1/2]**
>
> We thank the reviewer for raising critical and insightful questions regarding computational cost, implementation details, results, and analysis. We first respond to the key issues related to computational cost, and then address the weaknesses and questions point by point below. In addition, we have uploaded a revised version of the paper to address these concerns.
>
> ----
>
> ## **Key issue: the computational efficiency comparison between MIL-based methods and CCFormer**
>
> We compare the end-to-end computational efficiency of MIL methods and CCFormer on WSI-level tasks. Specifically, UNI is used to extract patch features for the MIL methods, as described in **Section 5.1**, and DPA-P2PNet is used for cell detection, as detailed in **Appendix B.1**. All evaluations are conducted on NVIDIA H20 GPUs.
>
> ### **Preprocessing**
>
> The comparison between UNI and DPA-P2PNet is summarized in the table below. For UNI, patches are resized to 224, which is the default input size. For DPA-P2PNet, the default input size is 512, as detailed in **Appendix B.1**. The WSI-level results are averaged over about 1,000 WSIs.
>
> | Setting           | Method     | Param. | Patch size | Throughput (patches / s) | FLOPs (per patch) | Runtime (per WSI, s)* | FLOPs (per WSI) |
> |------------------|-----------|--------|------------|---------------------------|-------------------|----------------------|-----------------|
> | Default          | UNI       | 303M   | 224        | 278                       | 60 GFLOPs         | 41                   | 672 TFLOPs      |
> | Default          | DPA-P2PNet| 55M    | 512        | 160                       | 87 GFLOPs         | 70                   | 981 TFLOPs      |
> | Inference in 20x | DPA-P2PNet| 55M    | 256        | 612                       | 22 GFLOPs         | 18                   | 245 TFLOPs      |
> | Better FM        | UNI v2    | 681M   | 224        | 114                       | 180 GFLOPs        | 99                   | 2.0 PFLOPs      |
>
> $*$ Appendix B.4 reports the per-WSI processing time, which additionally includes common I/O procedures such as WSI patching, data loading, and result saving. Here, we consider only the computation time to enable a fairer comparison.
>
> Compared with DPA-P2PNet, UNI has 5.5 times more parameters, which substantially increases the computational cost of MIL-based methods in the preprocessing stage. Although the default input patch size of DPA-P2PNet is 2.3 times that of UNI, the inference cost per patch remains comparable between UNI and DPA-P2PNet.
>
> In addition, the input size for cell detection and classification can be reduced from 512 at 40× magnification to 256 at 20× magnification to further improve computational efficiency. Our computational analysis shows that, under this setting, the average WSI inference time based on DPA-P2PNet is only 0.44× that of UNI. To assess the impact of this change on cell detection performance at 40× and 20×, we conduct an analysis on the Lizard dataset. Specifically, DPA-P2PNet achieves F1 scores of 88.08 and 87.90 at 40× and 20× magnification, respectively. This shows that performing cell inference at a lower magnification can substantially accelerate cell cloud inference without compromising accuracy.
>
> However, more advanced foundation models further increase the number of parameters, imposing a substantial computational burden on MIL-based methods. For example, UNI V2 contains 681M parameters. The inference cost of UNI V2 for a single WSI increases to 2.0 PFLOPs.

---

> ### Author Response · Authors · 2025-11-25
> **Key issue: the computational efficiency comparison [2/2]**
>
> ### **WSI-level tasks**
>
> We further compare the inference computational cost of MIL-based methods and CCFormer on WSI-level tasks. The runtime is measured as the average forward-pass time with a batch size of 1. Both runtime and FLOPs exclude preprocessing and data loading.
>
> | Method           | Memory (GB) | Runtime (per WSI, s) | FLOPs (per WSI) |
> |------------------|-------------|----------------------|-----------------|
> | ABMIL            | 2.4         | 0.0012               | 9 GFLOPs        |
> | TransMIL         | 9.3         | 0.0116               | 63 GFLOPs       |
> | PointNet         | 2.7         | 0.0201               | 17 GFLOPs       |
> | CCFormer (ours)  | 67          | 0.2050               | 117 GFLOPs      |
>
> PointNet exhibits memory usage comparable to MIL-based methods. The FLOPs per WSI of PointNet are also significantly lower than those of TransMIL. Moreover, PointNet achieves substantially better performance than both ABMIL and TransMIL. These results indicate that representing WSIs as cell clouds is an effective approach that does not incur higher computational costs than MIL-based methods.
>
> Although CCFormer requires more GPU memory, longer runtime, and higher FLOPs than MIL-based methods that only perform feature aggregation, it can still infer WSIs within one second on a single GPU with at least 80 GB of memory. Moreover, CCFormer offers several advantages: 1) it explicitly models the cell spatial distribution of WSIs rather than merely aggregating patch features, and 2) it achieves significantly better survival prediction performance across multiple cancer types compared to MIL-based methods.
>
> ### **An analysis from the perspective of information compression**
>
> We would like to further analyze the computational complexity of MIL-based methods and cell cloud-based methods from the perspective of information compression. Since a WSI at 40× magnification typically contains hundreds of millions of pixels, end-to-end modeling poses a substantial challenge. We argue that MIL-based methods and cell cloud-based methods represent two different information compression strategies that enable WSIs to be analyzed with limited computational resources.
>
> For MIL-based methods, we employ UNI to compress a patch at 40× magnification of size [512, 512, 3] into a feature vector of size [1024, 1], resulting in a new representation whose memory usage is approximately 0.13% of the original data.
>
> For the same patch at 40× magnification of size [512, 512, 3], the average number of detected cells is only 52. If we use the x-coordinate, y-coordinate, and type to describe each cell, the original RGB patch is compressed into a tensor of size [52, 3], whose memory usage is only about 0.02% of the original data.
>
> Therefore, the WSI representation based on cell clouds is more memory-efficient than patch-based methods. For WSI-level tasks, this enables us to design more complex models on the same hardware to learn the spatial distribution of cells, rather than only training an aggregator.

---

> ### Author Response · Authors · 2025-11-25
> **Response to Question 1**
>
> ## **1. There is inevitably a significant computational cost for segmentation, which largely hinders real-world applications.**
>
> We appreciate the reviewer for raising this critical issue that affects the applicability of CCFormer to real‑world problems. As shown in **the response to the Key Issue**, compared with UNI, which extracts features from all patches in a WSI, performing cell detection over the whole WSI does not introduce a significant increase in computational cost and runtime.
>
> More importantly, we further clarify that performing cell detection at 20× magnification can substantially reduce the average preprocessing FLOPs for each WSI without sacrificing detection accuracy. In contrast, for MIL-based methods, using a more powerful foundation model such as UNI V2 for feature extraction would increase the average preprocessing FLOPs per WSI to 2 PFLOPs.
>
> Therefore, the pipeline of cell detection + CCFormer does not introduce a significantly higher preprocessing cost compared with the pipeline of feature extractor + MIL-based methods. In addition, cell detection models, similar to general object detection models, have strong potential for extremely lightweight designs that can further reduce computational cost. We will explore such models in future work to minimize the preprocessing cost of CCFormer as much as possible.
>
> In the revised version, we have added **Appendix E** to provide a detailed comparison of computational efficiency.

---

> ### Author Response · Authors · 2025-11-25
> **Response to Question 2**
>
> ## **2. Please elaborate on the usage of mean aggregation and maximum aggregation.**
>
> We thank the reviewer for their attention to the implementation details of CCFormer. We would like to elaborate on the design of aggregation in CCFormer and FusedCCFormer, respectively.
>
> ### **CCFormer**
>
> At the low levels, in order to provide sufficiently rich and stable representations for subsequent layers, low-level statistical characteristics, such as the proportions and spatial distributions of different cell types within local regions, need to be preserved. If maximum aggregation is used at these levels, only the strongest response within each local region would be retained, causing substantial information to be discarded prematurely.
>
> By contrast, at the final layer, CCFormer has already integrated local information multiple times and produced higher-level semantic features. Using maximum aggregation allows each dimension to retain only the strongest response across the entire slide, thereby emphasizing the presence of strongly activated, potentially critical structures and enhancing the discriminative of CCFormer for slide-level tasks. In addition, we would like to emphasize that maximum aggregation is also the default last-level aggregation method in point cloud models such as PointNet and PointMLP.
>
> To further validate the design, we conduct a comprehensive ablation study on the aggregation methods.
>
> | Aggregation of low levels | Aggregation of the last level | C-Index of PAAD |
> |---------------------------|-------------------------------|-----------------|
> | Maximum Aggregation       | Maximum Aggregation           | 0.705 ± 0.085   |
> | Maximum Aggregation       | Mean Aggregation              | 0.719 ± 0.075   |
> | Mean Aggregation          | Mean Aggregation              | 0.731 ± 0.038   |
> | Mean Aggregation          | Maximum Aggregation           | 0.748 ± 0.056   |
>
> The results show that:
> 1) Compared with maximum aggregation, using mean aggregation in the low levels can significantly improve the performance of CCFormer;
> 2) Using maximum aggregation in the final layer yields better performance than mean aggregation.
>
> ### **FusedCCFormer**
>
> Different from cell clouds, we find that mean aggregation performs better for appearance representation. In particular, as shown in **Table 1**, MeanPool (patch) significantly outperforms MaxPool (patch). This observation motivates us to adopt mean aggregation for aggregating patch features in FusedCCFormer.
>
> | Method           | BLCA           | BRCA           | COAD READ       | LUAD           | PAAD           | STAD           |
> |------------------|----------------|----------------|-----------------|----------------|----------------|----------------|
> | MeanPool (Patch) | 0.610 ± 0.024  | 0.643 ± 0.031  | 0.629 ± 0.186   | 0.571 ± 0.057  | 0.672 ± 0.097  | 0.579 ± 0.085  |
> | MaxPool (Patch)  | 0.510 ± 0.038  | 0.589 ± 0.063  | 0.585 ± 0.078   | 0.480 ± 0.037  | 0.404 ± 0.122  | 0.474 ± 0.102  |

---

> ### Author Response · Authors · 2025-11-25
> **Response to Questions 3 and 4**
>
> ## **3. Some results are missing in Table 1 without any explanation**
>
> To compare with more advanced graph-based methods, we adopt PAMOE as a baseline. However, the released source code of PAMOE only includes the PAMOE+TransMIL implementation, so we initially reported the PAMOE+Patch-GCN results directly from their paper.
>
> Following the PAMOE supplementary material, we implement PAMOE+Patch-GCN and have updated the results for all cancer types in the revised version.
>
> | Method              | BLCA            | BRCA            | COADREAD        | LUAD            | PAAD            | STAD            |
> |---------------------|-----------------|-----------------|------------------|-----------------|-----------------|-----------------|
> | Patch-GCN           | 0.597 ± 0.022   | 0.628 ± 0.036   | 0.634 ± 0.121    | 0.617 ± 0.043   | 0.668 ± 0.115   | 0.563 ± 0.048   |
> | PAMOE+Patch-GCN     | 0.593 ± 0.028   | 0.589 ± 0.026   | 0.643 ± 0.086    | 0.620 ± 0.049   | 0.640 ± 0.079   | 0.549 ± 0.046   |
> | CCFormer (ours)     | 0.641 ± 0.019   | 0.696 ± 0.056   | 0.782 ± 0.071    | 0.661 ± 0.034   | 0.748 ± 0.056   | 0.695 ± 0.076   |
> | FusedCCFormer (ours)| 0.667 ± 0.023   | 0.729 ± 0.066   | 0.795 ± 0.068    | 0.672 ± 0.045   | 0.771 ± 0.072   | 0.702 ± 0.063   |
>
> The reproduced PAMOE+PatchGCN performance on LUAD and PAAD is similar to the results reported in the original paper. Our reproduced PAMOE+PatchGCN results are also consistent with the conclusions reported in the original paper (Section 5.3 of PAMOE) that PAMOE does not consistently improve PatchGCN performance across all cancer types.
>
> In addition, since the code for Pixel-Mamba has not been released and its pipeline involves complex pre-training and end-to-end training, we still directly report the results from the original paper.
>
> ----
>
> ## **4. Did the authors borrow results from existing works?**
>
> To ensure fair comparisons, we reproduced the results of all baselines on all cancer types using the same split files and the officially released code, rather than directly borrowing the results from their papers. In the revised version, we additionally reproduced PAMOE+PatchGCN.
>
> We thank the reviewer for raising this concern. In the revised version, we explicitly clarify in **Section 5.1** that all baseline results are reproduced using the same split files and the released code of the baseline methods, unless otherwise specified. For Pixel-Mamba, we clearly indicate in an additional note under **Table 1** that its results are directly taken from the original paper.
>
> ----
>
> ## **5. LGG dataset**
>
> This work is designed to be applied to as many types of cancer as possible. Therefore, we categorize cells into three basic types: neoplastic, inflammatory, and other. Experiments on survival prediction and cancer staging validate the effectiveness of this design.
>
> This design also limits the performance of cell cloud-based methods for certain cancers that require fine-grained cell information. In particular, clinical studies have shown that the diagnosis of Lower Grade Gliomas (LGG) is highly dependent on detailed features such as mitotic figures, nuclear atypia, necrosis, and microvascular proliferation [1]. However, the current cell cloud design cannot effectively capture these fine-grained features. Therefore, experiments on LGG are beyond the scope of the present study.
>
> We thank the reviewer for raising the question regarding LGG. In **Appendix C**, we have further expanded the discussion about limitations. Specifically, we provide additional analyses for cancer types that are not included in the main experiments, such as KIRC and LGG. In future work, we plan to explore approaches to mitigate these issues, for example, by incorporating cell contour descriptors and performing further subclassification of cell types.
>
> ----
>
> [1] Eleonora Duregon, Luca Bertero, Alessandra Pittaro, Riccardo Soffietti, Roberta Ruda, Morena Trevisan, Mauro Papotti, Laura Ventura, Rebecca Senetta, and Paola Cassoni. Ki-67 proliferation index but not mitotic thresholds integrates the molecular prognostic stratification of lower grade gliomas. Oncotarget, 7(16):21190, 2016.

---

> ### Author Response · Authors · 2025-11-25
> **Response to Questions 5**
>
> ## **6. The results of ACC and AUC for cancer staging**
>
> We thank the reviewer for raising this concern. In the revised version, we have added the results of ACC and AUC to **Table 5**. The new results also demonstrate that CCFormer achieves SOTA performance in most metrics.
>
> ----
>
> ## **7. Please specify whether the improvement in the percentage format is in absolute or relative terms.**
>
> We thank the reviewer for raising this question, which helps clarify the discussion of our results. All reported results in percentages refer to relative improvements, computed as: (performance gain / baseline performance) x 100%.  In the revised version, we have explicitly stated in **Section 5.1** that the discussion of results is mainly based on relative improvements.
>
> ----
>
> We thank the reviewer again for the constructive feedback that help strengthen the comprehensiveness of the study. We hope that these revisions, additional experiments, and clarifications adequately address the reviewer’s concerns.

---

### Official Review · Reviewer_4BB4 · 2025-10-31

**Soundness:** 3
**Presentation:** 2
**Contribution:** 3
**Rating:** 4
**Confidence:** 3

**Summary:**

This paper proposes treating WSIs as cell point clouds, a new paradigm distinct from MIL. Specifically, the paper introduces a human-in-the-loop label-refinement method to tackle domain shift between the pretraining dataset (PanNuke) and the target dataset (the TCGA series), and a cell-cloud transformer to model cell spatial distribution. Extensive experiments demonstrate the effectiveness of this method.

**Strengths:**

1.	The paper is generally well-organized and easy to follow, and would be better if the methodology section could easier to understand.

2.	The main idea of treating WSIs as cell point clouds is innovative and aligns well with clinical practice.

3.	The experimental results demonstrate the effectiveness of this method along with its components.

**Weaknesses:**

1.	The choice of PanNuke is not entirely convincing. In Appendix C, the authors state that PanNuke is chosen due to the distribution similarities between PanNuke and TCGA. What about merging other segmentation datasets, such as MoNuSAC and Lizard, since more data generally leads better generalizability?

2.	The authors are advised to disclose the foundation model’s precision and recall (or related metrics) used during HLLR, which would help solidify the effectiveness of the approach.

3.	There should be more baselines for comparison in the main experiments: for graphs, HEAT (CVPR) and the Integrative Graph-Transformer framework (MICCAI); for patch-feature MIL, HDMIL (CVPR) and R2T-MIL (CVPR).

4.	Formatting issues: \citep{xxx} should be used in most cases, while the authors use \citet{xxx}. Define CCFormer when it first appears in the main text. In addition, what is the difference between hierarchical CCFormer and CCFormer? If they are the same, please use the terminology consistently.

**Questions:**

In lines 156 to 158, were these experiments conducted by the authors, or are they results from previous studies?

---

> ### Author Response · Authors · 2025-11-25
> **Response to Weaknesses 1 and 2**
>
> We appreciate the reviewer for the very valuable comments and suggestions regarding HLLR, baselines, formatting, and results. We respond to weaknesses and questions point by point, and have uploaded a revised version of the paper that incorporates these clarifications and improvements.
>
> ----
>
> ## **1. The choice of PanNuke is not entirely convincing**
>
> We appreciate the reviewer for raising this important concern, and we would like to more thoroughly explain our choice of using PanNuke for training.
>
> Beyond the distributional similarity between PanNuke and TCGA, PanNuke has a particularly crucial advantage: it covers 19 different organs. This broad coverage allows us to train a cell detection and classification model that performs well across a wide range of tissues, which is essential for generalizable applications and for supporting downstream tasks on multiple cancer types.
>
> In contrast, other commonly used datasets cover only a limited subset of organs. For example, MoNuSAC includes only 4 organs (lung, prostate, kidney, and breast), and Lizard focuses solely on colon tissue. All of these are already encompassed within PanNuke. Therefore, PanNuke provides a substantially more diverse and representative training basis for building a cell detection and classification model.
>
> We have further clarified the justification for choosing PanNuke as the training dataset for cell detection and classification in **Appendix B.1** of the revised version.
>
> ----
>
> ## **2. What about merging other segmentation datasets?**
>
> We thank the reviewer for this insightful suggestion. First, we would like to emphasize that CCFormer does not rely on pixel-level segmentation. It only requires cell center (point) annotations and cell-type labels. This significantly reduces the annotation cost compared with dense segmentation masks.
>
> Our experiments with HLLR clearly indicate that introducing more annotated data indeed improves both cell detection and downstream slide-level performance (Section 5.3). Concretely, in HLLR we added 743 newly annotated samples (141,649 cells, **Appendix B.1**) to fine-tune the cell detection and classification model. This resulted in a 6% increase in the number of detected cells. Moreover, CCFormer and FusedCCFormer achieved 1.2% and 2.4% performance gains, respectively, on survival prediction.
>
> We thank the reviewer again for this valuable suggestion. We will try to incorporate more public datasets and annotated data to further improve the accuracy of cell detection and classification, thereby enhancing the performance of cell cloud–based models on slide-level tasks in our future work.
>
> ----
>
> ## **3. The foundation model’s metrics used during HLLR**
>
> We thank the reviewer for the suggestion that analyzing the performance of Foundation Models (FMs) during HLLR would help solidify the effectiveness of the approach.
>
> Specifically, 191 patches are additionally annotated by experts to evaluate patch-level classification accuracy. The results show that the FMs achieve an overall accuracy of 68.1%. Moreover, the subset of credible patches selected by HLLR reaches an accuracy of 95.2%. The results support the effectiveness of HLLR in filtering and leveraging high-confidence patches for domain-adaptation finetuning.

---

> ### Author Response · Authors · 2025-11-25
> **Response to Weaknesses 3 and 4, and Question 1**
>
> ## **4. More baselines for comparison in the main experiments**
>
> We thank the reviewer for the suggestion to include additional baselines and have added HDMIL and R$^2$T-MIL to the main experiments in the revised version.
>
> | Method            | BLCA              | BRCA              | COADREAD          | LUAD              | PAAD              | STAD              |
> |-------------------|-------------------|-------------------|--------------------|-------------------|-------------------|-------------------|
> | HDMIL             | 0.597 ± 0.070     | 0.605 ± 0.086     | 0.607 ± 0.029      | 0.620 ± 0.092     | 0.748 ± 0.075     | 0.621 ± 0.108     |
> | R$^2$T-MIL           | 0.648 ± 0.031     | 0.640 ± 0.103     | 0.697 ± 0.051      | 0.655 ± 0.101     | 0.712 ± 0.072     | 0.638 ± 0.117     |
> | CCFormer (ours)   | 0.641 ± 0.019     | 0.696 ± 0.056     | 0.782 ± 0.071      | 0.661 ± 0.034     | 0.748 ± 0.056     | 0.695 ± 0.076     |
> | FusedCCFormer (ours) | 0.667 ± 0.023 | 0.729 ± 0.066     | 0.795 ± 0.068      | 0.672 ± 0.045     | 0.771 ± 0.072     | 0.702 ± 0.063     |
>
> CCFormer still outperforms both HDMIL and R$^2$T-MIL.
>
> Since the Integrative Graph-Transformer framework (MICCAI) has not released its code, we are currently unable to conduct a direct comparison with it. When attempting to reproduce HEAT, we encountered the same issues reported in its official code repository and are still working hard to obtain a stable reimplementation. We will update our results as soon as we have successful experiments with HEAT.
>
> We appreciate the reviewer again for these valuable suggestions. In the revised version, we have updated the related work section to include citations to HEAT (CVPR), the Integrative Graph-Transformer framework (MICCAI), HDMIL (CVPR), and R2T-MIL (CVPR).
>
> -----
>
> ## **5. Formatting issues: \citep{xxx} and \cite{xxx}**
>
> We sincerely appreciate the reviewer for pointing out these formatting issues. We have refined the citation style throughout the paper to ensure the appropriate use of \citep and \citet.
>
> ----
>
> ## **6. Define CCFormer when it first appears in the main text**
>
> We thank the reviewer for raising this concern and have carefully rechecked the Introduction. CCFormer is clearly defined at its first appearance in both **the abstract (Line 21)** and **the main text (Line 84)**. In the revised version, we provide an explicit definition of CCFormer in **Figure 1** to further improve clarity.
>
> ----
>
> ## **7. The difference between hierarchical CCFormer and CCFormer**
>
> We thank the reviewer for raising this concern and apologize for the inconsistency in terminology. **Hierarchical CCFormer** and **CCFormer** refer to the same model. The term **hierarchical CCFormer** is only used in **the Introduction (Line 89 and Line 100)** to emphasize the hierarchical modeling of cell clouds, while all other parts of the paper consistently use **CCFormer**. In the revised version, we have unified the terminology by replacing **hierarchical CCFormer** with **CCFormer** and only using the full name **hierarchical Cell Cloud Transformer** when introducing the model.
>
> ----
>
> ## **8. Were these experiments (lines 156 to 158) conducted by the authors ?**
>
> We thank the reviewer for raising this concern. We design and conduct the pilot study. In **Appendix B.2**, we provide more detailed information about its setup and analysis.
>
> Through this pilot study, we demonstrate that simple cell-level statistics can already be used to develop effective clinical indicators, which provides strong empirical support for cell cloud–based WSI representations. These findings further motivated us to develop CCFormer to model cell spatial distributions at scale.
>
> ----
>
> We thank the reviewer again for detailed suggestions on both the experiments and the formatting, which help further improve the presentation of this work. We hope that the revisions, additional clarifications, and the more thorough formatting checks and refinements can adequately address the reviewer’s concerns.

---

### Official Review · Reviewer_DM3L · 2025-11-01

**Soundness:** 2
**Presentation:** 2
**Contribution:** 2
**Rating:** 4
**Confidence:** 5

**Summary:**

This paper presents a novel paradigm for WSI analysis by modeling them as "cell clouds" instead of image patches. The authors propose a two-stage approach: first, they convert WSIs into cell point clouds using a cell detection model fine-tuned with a Human-in-the-Loop strategy; second, they introduce a hierarchical transformer, CCFormer, to learn spatial distributions from these clouds for downstream tasks. Experiments on survival prediction and cancer staging across TCGA datasets show that the method achieves state-of-the-art performance, particularly when fused with appearance features.

**Strengths:**

The paper's primary strength is its novel formulation of the WSI analysis problem. Shifting the focus from conventional image patches to a holistic "cell cloud" representation is an original and interesting direction. This cell-based perspective holds the potential for better model interpretability by directly linking predictions to cellular spatial patterns, a significant advantage over abstract patch-based features. The technical approach is also reasonably well-designed; the proposed Neighboring Information Embedding (NIE) and Hierarchical Spatial Perception (HSP) modules are logical and well-motivated components for capturing local and global cell distribution characteristics within this new framework.

**Weaknesses:**

Despite its novel perspective, the paper suffers from several significant weaknesses that undermine the reproducibility, experimental depth, and the broader impact of its claims.

*   **W1: Unsubstantiated Claims of Cost-Efficiency.** The paper repeatedly suggests its approach is efficient and low-cost, but fails to provide the necessary evidence or context.
    *   **Ambiguous Annotation Cost:** The Human-in-the-Loop Label Refinement (HLLR) is presented as a "low-cost" solution. However, this claim is made in a vacuum. The paper neglects to discuss that the primary competitors, weakly-supervised MIL methods, require only inexpensive slide-level labels, whereas HLLR requires a subset of highly granular (and thus expensive) cell-level annotations. To be convincing, the authors must provide a direct discussion of this trade-off, ideally quantifying the expert time required and justifying why this higher-quality annotation effort is more cost-effective than standard weak supervision.
    *   **Inadequate Computational Cost Analysis:** The claim of 2.1 minutes/WSI on a high-end NVIDIA H20 GPU is not a meaningful benchmark without a direct comparison to the main baselines. The paper is missing a critical experiment: an end-to-end timing comparison on the *same hardware* against a representative patch-based method (e.g., UNI feature extraction + MIL aggregator).

*   **W2: Critical Lack of Evaluation for Upstream and Downstream Components.** The paper's evaluation is incomplete, making it difficult to understand the source of performance gains and the model's behavior.
    *   **Missing Cell Classifier Performance:** The methodology's success hinges on the quality of the initial cell cloud, yet the performance of the cell detection and classification model is never reported (e.g., F1-score, Precision). Without this, it is impossible to disentangle the contributions of the cell classifier from the CCFormer architecture, hindering error analysis and reproducibility.
    *   **Insufficient Ablation for FusedCCFormer:** The FusedCCFormer model shows significant performance gains, suggesting that patch-based appearance features are crucial. However, the paper lacks a proper ablation study. The authors should present results for three distinct models under their framework: (1) CCFormer using only cell features, (2) a baseline using only the patch-based foundation model features (e.g., global average pooling of UNI features), and (3) the combined FusedCCFormer. This would clarify the complementary value of each feature type.

*   **W3: Lack of Clarity and In-depth Interpretation.** Key aspects of the methodology and results are not clearly explained.
    *   **Unclear Visualizations:** The interpretation of several figures is opaque. For instance, the conclusion drawn from the similarity map in Figure 6(c) is not well-supported by the provided explanation. It is unclear how the visualization demonstrates that the model "comprehends semantic relationships" beyond simply showing an attention map. A more rigorous explanation is needed.
    *   **Missed Opportunity for Interpretability Analysis:** A primary motivation for cell-based analysis is the potential for high-level interpretability—linking patient outcomes to specific cellular compositions and spatial arrangements (e.g., tumor-immune interactions). Despite claiming this as an advantage, the paper does not provide any in-depth analysis of the learned representations to offer such clinical or biological insights. This is a significant missed opportunity to showcase the most compelling advantage of their paradigm over less interpretable patch-based methods. Given that patch-based approaches likely have an edge in annotation and computational cost, a strong demonstration of interpretability is crucial to justify the proposed framework.

*   **W4: Limited Practicality due to Hardware Specificity.** The experiments were conducted on an NVIDIA H20, a specialized datacenter GPU. The lack of benchmarks on more common hardware (e.g., consumer-grade RTX series) makes it difficult to assess the practical applicability and accessibility of the method.

**Questions:**

Thank you for this interesting work. I have several questions that I hope you can address during the rebuttal phase. My assessment of the paper could change significantly based on your responses.

1.  **Regarding Cost-Benefit Analysis:** Your work's central premise is a shift towards a new paradigm. To evaluate this shift, a clear understanding of its costs is crucial.
    *   **Q1.1 (Annotation Cost):** Could you please provide an estimate of the expert annotation time (in hours) required to label the 743 samples for the HLLR fine-tuning? More importantly, could you elaborate on why this targeted, cell-level annotation is more cost-effective than using readily available slide-level labels required by standard weakly-supervised MIL methods?
    *   **Q1.2 (Computational Cost):** Could you provide an end-to-end computational time comparison between your full pipeline (cell detection + CCFormer) and a leading patch-based baseline (e.g., UNI feature extraction + MambaMIL) on the same hardware for a representative WSI? This would provide a much-needed direct comparison of efficiency.

2.  **Regarding Component Evaluation & Ablation:** The contribution of each part of your pipeline is currently unclear.
    *   **Q2.1 (Cell Classifier Performance):** What was the performance (e.g., F1-score for each class) of your fine-tuned DPA-P2PNet cell classifier on a held-out test set? Providing these metrics is critical for understanding the quality of the input to CCFormer and for assessing the reproducibility of your results.
    *   **Q2.2 (Feature Ablation):** The performance leap of FusedCCFormer is substantial. Could you provide ablation results that compare three models: (1) CCFormer (cell features only), (2) a baseline using UNI features only (e.g., with global average pooling), and (3) FusedCCFormer? This would precisely quantify the individual and complementary contributions of cell-spatial and image-appearance features.

3.  **Regarding Interpretability:** The potential for interpretability is a key advantage of your cell-based approach, but it remains underexplored.
    *   **Q3.1 (Clarification of Figure 6c):** Could you please provide a more detailed explanation of how the attention map in Figure 6(c) demonstrates the comprehension of "semantic relationships"? Specifically, what are the semantic characteristics of Region I, II, and III that lead to these high/low attention scores?
    *   **Q3.2 (Deeper Interpretability Analysis):** Beyond visualizations, have you performed any analysis to link specific learned cell spatial patterns (e.g., high density of immune cells near tumor clusters) to patient outcomes? Showcasing even one such concrete example would significantly strengthen the claim that your method offers superior interpretability over patch-based models, which is arguably the most important justification for the additional costs of your framework.

**Details Of Ethics Concerns:**

N/A.

---

> ### Author Response · Authors · 2025-11-25
> **Key issue: the computational efficiency comparison [1/2]**
>
> ## **Key issue: the computational efficiency comparison between MIL-based methods and CCFormer**
>
> We compare the end-to-end computational efficiency of MIL methods and CCFormer on WSI-level tasks. Specifically, UNI is used to extract patch features for the MIL methods, as described in Section 5.1, and DPA-P2PNet is used for cell detection, as detailed in Appendix B.1. All evaluations are conducted on NVIDIA H20 GPUs.
>
> ### **Preprocessing**
>
> The comparison between UNI and DPA-P2PNet is summarized in the table below. For UNI, patches are resized to 224, which is the default input size. For DPA-P2PNet, the default input size is 512, as detailed in Appendix B.1. The WSI-level results are averaged over about 1,000 WSIs.
>
> | Setting           | Method     | Param. | Patch size | Throughput (patches / s) | FLOPs (per patch) | Runtime (per WSI, s)* | FLOPs (per WSI) |
> |------------------|-----------|--------|------------|---------------------------|-------------------|----------------------|-----------------|
> | Default          | UNI       | 303M   | 224        | 278                       | 60 GFLOPs         | 41                   | 672 TFLOPs      |
> | Default          | DPA-P2PNet| 55M    | 512        | 160                       | 87 GFLOPs         | 70                   | 981 TFLOPs      |
> | Inference in 20x | DPA-P2PNet| 55M    | 256        | 612                       | 22 GFLOPs         | 18                   | 245 TFLOPs      |
> | Better FM        | UNI v2    | 681M   | 224        | 114                       | 180 GFLOPs        | 99                   | 2.0 PFLOPs      |
>
> $*$ Appendix B.4 reports the per-WSI processing time, which additionally includes common I/O procedures such as WSI patching, data loading, and result saving. Here, we consider only the computation time to enable a fairer comparison.
>
> Compared with DPA-P2PNet, UNI has 5.5 times more parameters, which substantially increases the computational cost of MIL-based methods in the preprocessing stage. Although the default input patch size of DPA-P2PNet is 2.3 times that of UNI, the inference cost per patch remains comparable between UNI and DPA-P2PNet.
>
> In addition, the input size for cell detection and classification can be reduced from 512 at 40× magnification to 256 at 20× magnification to further improve computational efficiency. Our computational analysis shows that, under this setting, the average WSI inference time based on DPA-P2PNet is only 0.44× that of UNI. To assess the impact of this change on cell detection performance at 40× and 20×, we conduct an analysis on the Lizard dataset. Specifically, DPA-P2PNet achieves F1 scores of 88.08 and 87.90 at 40× and 20× magnification, respectively. This shows that performing cell inference at a lower magnification can substantially accelerate cell cloud inference without compromising accuracy.
>
> However, more advanced foundation models further increase the number of parameters, imposing a substantial computational burden on MIL-based methods. For example, UNI V2 contains 681M parameters. The inference cost of UNI V2 for a single WSI increases to 2.0 PFLOPs.

---

> ### Author Response · Authors · 2025-11-25
> **key issue: the computational efficiency comparison [2/2]**
>
> ### **WSI-level tasks**
>
> We further compare the inference computational cost of MIL-based methods and CCFormer on WSI-level tasks. The runtime is measured as the average forward-pass time with a batch size of 1. Both runtime and FLOPs exclude preprocessing and data loading.
>
> | Method           | Memory (GB) | Runtime (per WSI, s) | FLOPs (per WSI) |
> |------------------|-------------|----------------------|-----------------|
> | ABMIL            | 2.4         | 0.0012               | 9 GFLOPs        |
> | TransMIL         | 9.3         | 0.0116               | 63 GFLOPs       |
> | PointNet         | 2.7         | 0.0201               | 17 GFLOPs       |
> | CCFormer (ours)  | 67          | 0.2050               | 117 GFLOPs      |
>
> PointNet exhibits memory usage comparable to MIL-based methods. The FLOPs per WSI of PointNet are also significantly lower than those of TransMIL. Moreover, PointNet achieves substantially better performance than both ABMIL and TransMIL. These results indicate that representing WSIs as cell clouds is an effective approach that does not incur higher computational costs than MIL-based methods.
>
> Although CCFormer requires more GPU memory, longer runtime, and higher FLOPs than MIL-based methods that only perform feature aggregation, it can still infer WSIs within one second on a single GPU with at least 80 GB of memory. Moreover, CCFormer offers several advantages: 1) it explicitly models the cell spatial distribution of WSIs rather than merely aggregating patch features, and 2) it achieves significantly better survival prediction performance across multiple cancer types compared to MIL-based methods.
>
> ### **An analysis from the perspective of information compression**
>
> We would like to further analyze the computational complexity of MIL-based methods and cell cloud-based methods from the perspective of information compression. Due to the fact that a WSI at 40× magnification typically contains hundreds of millions of pixels, end-to-end modeling poses a substantial challenge. We argue that MIL-based methods and cell cloud-based methods represent two different information compression strategies that enable WSIs to be analyzed with limited computational resources.
>
> For MIL-based methods, we employ UNI to compress a patch at 40× magnification of size [512, 512, 3] into a feature vector of size [1024, 1], resulting in a new representation whose memory usage is approximately 0.13% of the original data.
>
> For the same patch at 40× magnification of size [512, 512, 3], the average number of detected cells is only 52. If we use the x-coordinate, y-coordinate, and type to describe each cell, the original RGB patch is compressed into a tensor of size [52, 3], whose memory usage is only about 0.02% of the original data.
>
> Therefore, the WSI representation based on cell clouds is more memory-efficient than patch-based methods. For WSI-level tasks, this enables us to design more complex models on the same hardware to learn the spatial distribution of cells, rather than only training an aggregator.

---

> ### Author Response · Authors · 2025-11-25
> **Response to Unsubstantiated Claims of Cost-Efficiency and Cost-Benefit Analysis**
>
> ## **1. Comparison of HLLR and weakly-supervised MIL methods**
>
> We thank the reviewer for raising this concern. However, there may be a misunderstanding. HLLR is not a WSI analysis method, but rather a strategy for improving cell detection accuracy. Therefore, it is not directly comparable to MIL-based methods, which are designed for slide-level representation and prediction. In **Section 5.3**, the ablation studies verify the effectiveness of HLLR. Furthermore, **Table 1** and **Figure 5** clearly show that CCFormer outperforms MIL-based methods.
>
> ----
>
> ## **2. Inadequate Computational Cost Analysis**
>
> We thank the reviewer for raising this critical question. As shown in **the response to the key issue**, cell cloud-based methods have a preprocessing computational cost that is comparable to that of MIL-based methods. For WSI-level tasks, PointNet and MIL-based methods exhibit similar memory usage and FLOPs, further demonstrating that a WSI representation based on cell clouds does not incur a higher computational cost than MIL-based approaches. Although CCFormer requires more GPU memory, longer runtime, and higher FLOPs than MIL-based methods, it can still infer WSIs within one second on a single GPU and outperforms MIL-based methods.

---

> ### Author Response · Authors · 2025-11-25
> **Response to Critical Lack of Evaluation and Ablation**
>
> ## **3. Missing Cell-level Performance**
>
> We thank the reviewer for raising this concern. **Section 5.3** presents the cell counts obtained before and after applying HLLR-based finetuning to the cell detection model. After incorporating HLLR into the finetuning process for cell detection and classification, the number of detected cells increases by roughly 6%. We also conducted an additional evaluation of the cell detection model on the PanNuke test set. Under this setting, the model achieves a precision of 86.72, a recall of 81.97, and an F1-score of 84.28.
>
> ----
>
> ## **4. Insufficient Ablation for FusedCCFormer**
>
> We thank the reviewer for raising this question. The three methods, 1) CCFormer using only cell features, 2) a baseline using only patch-based foundation model features, and 3) the combined FusedCCFormer, correspond to the penultimate row, the first row, and the last row of **Table 1**, respectively. The results show that CCFormer outperforms other SOTA methods. Furthermore, FusedCCFormer achieves even higher accuracy by fusing CCFormer with the appearance-based representation.
>
> | Method               | BLCA            | BRCA            | COADREAD        | LUAD            | PAAD            | STAD            |
> |----------------------|-----------------|-----------------|------------------|-----------------|-----------------|-----------------|
> | MeanPool (Patch)     | 0.610 ± 0.024   | 0.643 ± 0.031   | 0.629 ± 0.186    | 0.571 ± 0.057   | 0.672 ± 0.097   | 0.579 ± 0.085   |
> | CCFormer (ours)      | 0.641 ± 0.019   | 0.696 ± 0.056   | 0.782 ± 0.071    | 0.661 ± 0.034   | 0.748 ± 0.056   | 0.695 ± 0.076   |
> | FusedCCFormer (ours) | 0.667 ± 0.023   | 0.729 ± 0.066   | 0.795 ± 0.068    | 0.672 ± 0.045   | 0.771 ± 0.072   | 0.702 ± 0.063   |

---

> ### Author Response · Authors · 2025-11-25
> **Response to Lack of Clarity and In-depth Interpretation**
>
> ## **5. Unclear Visualizations**
>
> We thank the reviewer for raising this question and would like to provide a more detailed explanation of the visualization in **Figure 6(c)**. In **Figure 6(c)**, we provide a visualization of the similarity among the group features in the last layer of CCFormer. Specifically, we compute the pairwise similarity between the final-layer group features to characterize how the model organizes these regions in the embedding space. The visualization indicates that CCFormer captures meaningful semantic relationships among tissue regions: local regions with similar histological structures and cell spatial distribution exhibit high similarity, whereas regions with distinct tissue morphologies show low similarity. These findings suggest that CCFormer has learned high-level semantic representations after training, thereby verifying the high interpretability.
>
> ----
>
> ## **6. Missed Opportunity for Interpretability Analysis**
>
> We thank the reviewer for this insightful suggestion. We agree that interpretability is an important advantage of cell cloud–based WSI analysis. In this work, we have already conducted several forms of interpretability‑oriented analysis.
>
> First, in **Section 3**, we construct survival risk evaluation metrics based on cell clouds and perform Kaplan–Meier analyses. The results show that simple cell‑level statistics can serve as effective clinical indicators for stratifying patients.
>
> Second, in **Figure 6(c)**, we visualize the similarity between the group features in the last layer of CCFormer. The results indicate that CCFormer captures meaningful semantic relationships between regions: local tissue regions with similar structures exhibit high similarity in the learned representations, which supports the interpretability of the model at both tissue and cell levels.
>
> More complex and fine‑grained interpretability analyses, such as the relationship between specific cell cloud spatial patterns and patient risk, constitute an important direction that we plan to explore in future work.

---

> ### Author Response · Authors · 2025-11-25
> **Response to Limited Practicality due to Hardware Specificity**
>
> We thank the reviewer for this comment. Although experiments on multiple platforms and hardware configurations would further elucidate the cross-platform usability of CCFormer, the primary goal of this work is to introduce a novel cell cloud-based method for WSI analysis, rather than providing an exhaustive survey of hardware environments. Extensive experiments on survival prediction and cancer staging show that CCFormer achieves SOTA performances and evidently outperforms other competing methods by learning from cell spatial distribution alone.
>
> Since CCFormer is implemented using PyTorch and does not exploit H20-specific capabilities, it is possible to train and infer CCFormer with other GPUs. A systematic comparison across multiple hardware platforms is an interesting direction for future work, but it is beyond the scope of the present work.

---

### Official Review · Reviewer_3CZF · 2025-11-02

**Soundness:** 3
**Presentation:** 2
**Contribution:** 2
**Rating:** 4
**Confidence:** 3

**Summary:**

This paper aims to address the problem of WSI analysis by introducing the spatial distribution of cells as a key marker, which has not been explored in previous work. Specifically, in the proposed framework, the authors treat the cells in a WSI as a point cloud. There are two major components in the proposed model: (1) an active-learning–based expert annotation strategy to reduce labeling workload, and (2) a hierarchical transformer architecture that extracts cell distribution information at different scales. The model is evaluated on several datasets, and the experimental results demonstrate that the proposed method outperforms prior approaches on two tasks: survival prediction and cancer staging.

**Strengths:**

* The idea of introducing cell distribution and treating cells as a point cloud is interesting and reasonable, and has not been explored previously.

* The hierarchical approach to modeling the cell cloud is well-motivated and technically sound.

* Given the extremely high cell density in WSIs, the HLLR module offers a sensible solution to reduce annotation effort.

* The paper is well-written and easy to follow.

* The experiments are rigorous and comprehensive.

**Weaknesses:**

* Although the idea of modeling cell distribution for WSI analysis is novel, it is not very feasible in practical applications. WSIs are extremely high-resolution, and the high cell density results in a substantial annotation burden—even with an active-learning strategy. In addition, the computational complexity grows significantly with cell density. However, the paper does not provide any runtime or memory usage metrics for training or inference, nor does it compare the computational cost with existing models.

* CCFormer relies heavily on the performance of cell detection, yet the paper does not report any quantitative results (e.g., accuracy, precision) for the detection step, nor does it analyze how detection quality affects overall slide-level performance.

* CCFormer utilizes only cell spatial distribution and cell type for WSI analysis, while ignoring other informative features such as texture, morphology, and overall tissue structure. These features are typically crucial in patch-based methods and are important in clinical pathology.

* Some claims are made without adequate explanation or justification. For example, the statement “Heavy reliance on the foundation models results in high computational costs and suboptimal performance” is not supported with evidence or comparison.

**Questions:**

* How robust of  WSI level analysis w.r.t. cell detection performance?
* How the computational complexity of the proposed method compared with other patch-level models.
* How sensitive of CCFormer to the choice of number of group anchors at different level and other hyper-parameters of the hierachical grouping ?

---

> ### Author Response · Authors · 2025-11-25
> **Key issue: the computational efficiency comparison [1/2]**
>
> We appreciate the reviewer for the insightful comments regarding cell detection performance, computational efficiency, and ablation studies. We would like first to address the key issue concerning the computational efficiency comparison between MIL-based methods and CCFormer, and then respond to the weaknesses and questions point by point below. In addition, we have uploaded a revised version that incorporates these improvements and additional analysis.
>
> ----
>
> ## **Key issue: the computational efficiency comparison between MIL-based methods and CCFormer**
>
> We compare the end-to-end computational efficiency of MIL methods and CCFormer on WSI-level tasks. Specifically, UNI is used to extract patch features for the MIL methods, as described in Section 5.1, and DPA-P2PNet is used for cell detection, as detailed in Appendix B.1. All evaluations are conducted on NVIDIA H20 GPUs.
>
> ### **Preprocessing**
>
> The comparison between UNI and DPA-P2PNet is summarized in the table below. For UNI, patches are resized to 224, which is the default input size. For DPA-P2PNet, the default input size is 512, as detailed in Appendix B.1. The WSI-level results are averaged over about 1,000 WSIs.
>
> | Setting           | Method     | Param. | Patch size | Throughput (patches / s) | FLOPs (per patch) | Runtime (per WSI, s)* | FLOPs (per WSI) |
> |------------------|-----------|--------|------------|---------------------------|-------------------|----------------------|-----------------|
> | Default          | UNI       | 303M   | 224        | 278                       | 60 GFLOPs         | 41                   | 672 TFLOPs      |
> | Default          | DPA-P2PNet| 55M    | 512        | 160                       | 87 GFLOPs         | 70                   | 981 TFLOPs      |
> | Inference in 20x | DPA-P2PNet| 55M    | 256        | 612                       | 22 GFLOPs         | 18                   | 245 TFLOPs      |
> | Better FM        | UNI v2    | 681M   | 224        | 114                       | 180 GFLOPs        | 99                   | 2.0 PFLOPs      |
>
> $*$ Appendix B.4 reports the per-WSI processing time, which additionally includes common I/O procedures such as WSI patching, data loading, and result saving. Here, we consider only the computation time to enable a fairer comparison.
>
> Compared with DPA-P2PNet, UNI has 5.5 times more parameters, which substantially increases the computational cost of MIL-based methods in the preprocessing stage. Although the default input patch size of DPA-P2PNet is 2.3 times that of UNI, the inference cost per patch remains comparable between UNI and DPA-P2PNet.
>
> In addition, the input size for cell detection and classification can be reduced from 512 at 40× magnification to 256 at 20× magnification to further improve computational efficiency. Our computational analysis shows that, under this setting, the average WSI inference time based on DPA-P2PNet is only 0.44× that of UNI. To assess the impact of this change on cell detection performance at 40× and 20×, we conduct an analysis on the Lizard dataset. Specifically, DPA-P2PNet achieves F1 scores of 88.08 and 87.90 at 40× and 20× magnification, respectively. This shows that performing cell inference at a lower magnification can substantially accelerate cell cloud inference without compromising accuracy.
>
> However, more advanced foundation models further increase the number of parameters, imposing a substantial computational burden on MIL-based methods. For example, UNI V2 contains 681M parameters. The inference cost of UNI V2 for a single WSI increases to 2.0 PFLOPs.

---

> ### Author Response · Authors · 2025-11-25
> **Key issue: the computational efficiency comparison [2/2]**
>
> ### **WSI-level tasks**
> We further compare the inference computational cost of MIL-based methods and CCFormer on WSI-level tasks. The runtime is measured as the average forward-pass time with a batch size of 1. Both runtime and FLOPs exclude preprocessing and data loading.
>
> | Method           | Memory (GB) | Runtime (per WSI, s) | FLOPs (per WSI) |
> |------------------|-------------|----------------------|-----------------|
> | ABMIL            | 2.4         | 0.0012               | 9 GFLOPs        |
> | TransMIL         | 9.3         | 0.0116               | 63 GFLOPs       |
> | PointNet         | 2.7         | 0.0201               | 17 GFLOPs       |
> | CCFormer (ours)  | 67          | 0.2050               | 117 GFLOPs      |
>
> PointNet exhibits memory usage comparable to MIL-based methods. The FLOPs per WSI of PointNet are also significantly lower than those of TransMIL. Moreover, PointNet achieves substantially better performance than both ABMIL and TransMIL. These results indicate that representing WSIs as cell clouds is an effective approach that does not incur higher computational costs than MIL-based methods.
>
> Although CCFormer requires more GPU memory, longer runtime, and higher FLOPs than MIL-based methods that only perform feature aggregation, it can still infer WSIs within one second on a single GPU with at least 80 GB of memory. Moreover, CCFormer offers several advantages: 1) it explicitly models the cell spatial distribution of WSIs rather than merely aggregating patch features, and 2) it achieves significantly better survival prediction performance across multiple cancer types compared to MIL-based methods.
>
> ### **An analysis from the perspective of information compression**
>
> We would like to further analyze the computational complexity of MIL-based methods and cell cloud-based methods from the perspective of information compression. Since a WSI at 40× magnification typically contains hundreds of millions of pixels, end-to-end modeling poses a substantial challenge. We argue that MIL-based methods and cell cloud-based methods represent two different information compression strategies that enable WSIs to be analyzed with limited computational resources.
> For MIL-based methods, we employ UNI to compress a patch at 40× magnification of size [512, 512, 3] into a feature vector of size [1024, 1], resulting in a new representation whose memory usage is approximately 0.13% of the original data.
>
> For the same patch at 40× magnification of size [512, 512, 3], the average number of detected cells is only 52. If we use the x-coordinate, y-coordinate, and type to describe each cell, the original RGB patch is compressed into a tensor of size [52, 3], whose memory usage is only about 0.02% of the original data.
>
> Therefore, the WSI representation based on cell clouds is more memory-efficient than patch-based methods. For WSI-level tasks, this enables us to design more complex models on the same hardware to learn the spatial distribution of cells, rather than only training an aggregator.

---

> ### Author Response · Authors · 2025-11-25
> **Response to Weakness 1**
>
> ## **1. High cell density results in a substantial annotation burden**
>
> Cell clouds are generated with a pretrained cell detection and classification model (Appendix B.1), which is fully automatic and does not require manual annotations. HLLR and the expert-annotated data are used only to optimize the cell detection and classification model, rather than to optimize cell detection and classification predictions directly.
>
> We thank the reviewer for raising this concern, which prompted us to more clearly describe the preprocessing pipeline for WSIs. In particular, we have refined the description of HLLR and WSI preprocessing in Appendix B.1 in the revised version to avoid confusion regarding the cell detection and classification pipeline.
>
> ## **2. Runtime and memory usage metrics**
> We appreciate the reviewer for raising this question, which helps us further clarify the computational efficiency of CCFormer compared with other methods. In the revised version, we have added Appendix E to provide a thorough comparison of computational efficiency.
>
> ● **preprocessing**: The preprocessing cost comparison in Response 0 shows that feature extraction with FMs and cell detection with DPA-P2PNet have comparable computational costs under the default settings. In addition, we also show that performing cell inference at a lower magnification can substantially improve the efficiency of cell detection preprocessing while keeping the performance essentially unchanged. Therefore, modeling the cell spatial distribution based on cell detection and classification results does not introduce a significantly higher preprocessing burden compared with MIL-based methods.
>
> ● **WSI-tasks**: MIL-based methods only need to aggregate features extracted by FMs. As a result, they require less GPU memory, runtime, and FLOPs compared with CCFormer. However, this also limits the performance of MIL-based methods on WSI-level tasks. In contrast, CCFormer can fully capture cell-level relationships, model the spatial distribution of cells, and infer WSIs within one second on a single GPU. CCFormer does not rely on substantially higher hardware requirements and achieves SOTA performance on both survival prediction and cancer staging.

---

> ### Author Response · Authors · 2025-11-25
> **Response to Weakness 2**
>
> ## **3. Quantitative results of the detection step**
>
> In Section 5.3, we report both the original number of detected cells and the number obtained after finetuning the cell detection model with HLLR. The results show that, after finetuning the cell detection and classification model with HLLR, the model yields an approximately 6% increase in the number of detected cells. In addition, we further evaluated the cell detection model on the PanNuke test set. The results show that the precision, recall, and F1-score are 86.72, 81.97, and 84.28, respectively.
>
> We thank the reviewer for raising this concern. In the revised version, we have added Table 2 to present in detail the number of detected cells for each cancer dataset.
> | HLLR | BLCA | BRCA | COADREAD | LUAD | PAAD | STAD |
> |------|------|------|-----------|------|------|------|
> | ✗    | 449M | 623M | 396M      | 481M | 213M | 692M |
> | ✓    | 475M | 663M | 424M      | 510M | 226M | 736M |
>
> ## **4. The impact of detection quality on slide-level performance**
> We appreciate the reviewer for raising this critical question. As a key preprocessing step of CCFormer, the detection stage indeed affects the downstream slide-level performance. We conduct a detailed analysis of the impact of detection quality in our ablation studies (Line 423, Human-in-the-Loop Label Refinement, Section 5.3). In particular, our experiments show that applying HLLR to refine cell detection increases the number of cells by approximately 6%. Since each cancer dataset contains 200 to 700 million cells, tens of millions of additional cells per cancer type are detected, markedly strengthening the representation of cell spatial distribution. This improvement in cell detection quality leads to better performance in downstream survival prediction, where the average accuracy of CCFormer and FusedCCFormer increases by 1.5% and 2.4%, respectively.

---

> ### Author Response · Authors · 2025-11-25
> **Response to Weakness 3**
>
> ## **5. CCFormer ignores some informative features of WSIs**
>
> We fully agree that texture, morphology, and overall tissue structure are crucial for clinical pathology. However, our core claim is that WSIs can be represented by explicitly modeling the cell spatial distribution, an important pathological basis [1,2] that has long been overlooked by MIL-based methods. Clinical analysis and extensive experiments on downstream tasks support this claim and show that CCFormer achieves state-of-the-art performance.
>
> In addition, FusedCCFormer further fuses cell spatial distribution with appearance representations, leading to more comprehensive and expressive WSI representations.
>
> Therefore, we do not deny the importance of texture or other appearance features. Instead, we first demonstrate the importance of cell spatial distribution and then experimentally validate the effectiveness of fusing it with appearance representations.
>
> [1] Joel Saltz, Rajarsi Gupta, Le Hou, Tahsin Kurc, Pankaj Singh, Vu Nguyen, Dimitris Samaras, Kenneth R Shroyer, Tianhao Zhao, Rebecca Batiste, et al. Spatial organization and molecular correlation of tumor-infiltrating lymphocytes using deep learning on pathology images. Cell reports, 23(1):181–193, 2018.
>
> [2] German Corredor, Xiangxue Wang, Yu Zhou, Cheng Lu, Pingfu Fu, Konstantinos Syrigos, David L Rimm, Michael Yang, Eduardo Romero, Kurt A Schalper, et al. Spatial architecture and arrangement of tumor-infiltrating lymphocytes for predicting likelihood of recurrence in early-stage non–small cell lung cancer. Clinical cancer research, 25(5):1526–1534, 2019.

---

> ### Author Response · Authors · 2025-11-25
> **Response to Weakness 4**
>
> ## **6. Some claims are made without adequate explanation or justification**
>
> We appreciate the reviewer for raising this concern, and we would like to further clarify our statement that "heavy reliance on foundation models results in high computational costs and suboptimal performance." In the revised version, given that MIL-based methods still achieve excellent performance, we have removed the phrase "suboptimal performance" to more rigorously and conservatively describe the limitations of MIL-based methods.
>
> **High computational costs**: MIL requires using large foundation models to extract features for all patches, which imposes a heavy burden on the inference of each WSI. For example, as shown in **preprocessing analysis for the key issue of computational efficiency**, UNI requires on average 672 TFLOPs to process a single WSI. When using the more powerful UNI V2, the FLOPs per WSI further increase substantially to 2.0 PFLOPs. In contrast, the default cell detection cost for a single WSI in our pipeline is 981 TFLOPs, and after optimization it can be further reduced to 245 TFLOPs without sacrificing detection performance.

---

> ### Author Response · Authors · 2025-11-25
> **Response to Question 1**
>
> ## **7. How robust of WSI level analysis w.r.t. cell detection performance**
>
> We thank the reviewer for raising this important question. In **Section 5.3**, we provide a detailed analysis of the performance of cell cloud-based methods on downstream tasks after optimizing the cell detection model with HLLR. Specifically, HLLR increases the number of detected cells by approximately 6%, which in turn leads to performance improvements of 1.2% and 2.4% for CCFormer and FusedCCFormer, respectively, on the survival prediction.
>
> CCFormer achieves state-of-the-art performance based on cell clouds inferred by a model pretrained on public datasets, which demonstrates its strong robustness and good applicability to real-world problems. Moreover, better cell detection results can effectively enhance the performance of cell cloud-based methods on WSI-level tasks.

---

> ### Author Response · Authors · 2025-11-25
> **Response to Question 2**
>
> ## **8. How the computational complexity of the proposed method compared with other patch-level models.**
>
> We thank the reviewer for this question, which motivates us to clarify the computational cost of CCFormer more concretely. As discussed in **the response to the key issue**, cell cloud-based methods have a per-WSI average computational cost that is comparable to patch-level models. In addition, we further show that performing cell inference at 20× magnification can significantly reduce the computational complexity of cell detection. For WSI-level tasks, PointNet and TransMIL exhibit similar computational costs while achieving comparable performance. Although CCFormer has a higher computational complexity, its performance on WSI-level tasks is also substantially better, which we believe justifies the additional cost.

---

> ### Author Response · Authors · 2025-11-25
> **Response to Question 3**
>
> ## **9. Ablation studies of number of group anchors and other hyper-parameters of the hierachical grouping**
>
> We thank the reviewer for this suggestion, which helps us better validate the effectiveness of CCFormer. As shown in **Table 2**, we have conducted ablation studies on $N_{basic}$, which controls the number of group anchors at each level. The results show that CCFormer outperforms other SOTA methods across a wide range of $N_{basic}$ settings. Moreover, an appropriate N_basic further improves the modeling of local cell spatial distributions, enabling CCFormer to achieve a significantly higher performance than competing methods.
>
> We have also added ablation experiments on the number of group anchors in the first layer, $N_k$ in **Table 2**. The results show that CCFormer consistently achieves state-of-the-art performance across a wide range of $N_k$ settings.
>
> | $N_k$   | $N_{basic}$ | C-Index (↑)      |
> |------|--------|------------------|
> | 2048 | 4      | 0.652 ± 0.052    |
> | 2048 | 16     | 0.696 ± 0.056    |
> | 2048 | 32     | 0.662 ± 0.023    |
> | 1024 | 16     | 0.683 ± 0.045    |
> | 4096 | 16     | 0.663 ± 0.059    |
>
> ----
>
> We thank the reviewer again for these questions regarding computational efficiency, experiments, and cell detection performance, which help us further refine the comparison with MIL-based methods. We hope that the additional analyses of computational efficiency, together with the new results and corresponding discussions, can adequately address the reviewer’s concerns.

---

### Meta-Review · Area_Chair_Fnxr · 2025-12-19

**Summary:**

This work explores a new direction in WSI analysis, which treats the cells as point clouds for learning. Four reviewers have carefully reviewed this paper and provided valuable comments. All reviewers gave a rating of 4 (marginally below the acceptance threshold). The overall rating is below the acceptance bar.

The author wrote a comprehensive response in the rebuttal. In my view, most of the concerns have been well addressed. However, there are still some significant concerns remaining.

Reviewers 3CZF, DM3L, and aWo2 are all concerned about the computation costs, which I think is the major limitation of this method. As referred to the authors’ response: *“CCFormer requires more GPU memory, longer runtime, and higher FLOPs than MIL-based methods that only perform feature aggregation”*. Although I sincerely appreciate the authors’ honesty, the significantly increased computation costs may pose a major obstacle to the availability of your method, i.e., 10* runtime and 6* GFLOPs.

Another significant problem is the annotation burden. Reviewer DM3L raised a very valuable question, i.e., the annotation burden comparisons between HLLR and weakly supervised settings (MIL). I guess the authors misunderstood this question and did not adequately address it, as they answered, “HLLR is not a WSI analysis method, … it is not directly comparable to MIL-based methods”. Actually, the authors did not solve the concern of annotation costs. Considering HLLR is important to the cell detection task, as pointed out by the authors, I think this is not a trivial problem that can be ignored.

Reviewer DM3L also mentioned the interpretability analysis (linking patient outcomes to specific cellular compositions and spatial arrangements, e.g., tumor-immune interactions). The authors did not adequately address this part while leaving it to future work. But I think this can be solved within the rebuttal period, which is not a very heavy task that requires dense inputs.

In addition, I agree with the reviewers that there are some overclaims that are not adequately validated. The authors say that cell detection + CCFormer is better than the existing and widely-used workflows. It is a very strong claim and requires a more comprehensive evaluation to prove it. Considering the computation and annotation costs, I may not agree with this claim.

Although I commend the exploration of this interesting direction, I am sorry to recommend rejection based on the negative ratings of reviewers and the unresolved concerns.

**Reviewer Concerns:**

Some unresolved concerns have been listed as above.

**Reviewer Scores:**

Reviewer 4BB4 may improve the rating.

---

### Decision · Program_Chairs · 2026-01-26

Reject